# Node-Based Multiple Graph Learning with Theoretical Guarantees

## Abstract

In many applications, inferring graph topology, i.e., learning the graph structure from a given set of nodal observations, is a significant task. Existing approaches are mostly limited to learning a single graph assuming that the observed data are homogeneous. In many applications, data sets are heterogeneous and involve multiple related graphs, i.e., multiview graphs. Recent work on learning multiview graphs ensures the similarity of learned view graphs through edge-based similarity between the graphs. In this paper, we take a node-based approach instead of assuming that similarities and differences between networks are driven by individual edges, providing a more intuitive interpretation of network differences. Moreover, unlike existing methods that employ Gaussian Graphical Models (GGM), which learn precision matrices rather than the actual graph structures, we characterize the graph using a Laplacian matrix. Thus, the approach is expected to work broadly beyond Gaussian graphical learning. We develop an optimization framework to learn the individual graphical structures, assuming that the differences are due to individual nodes that are perturbed across views. The proposed optimization framework is presented for the special case of two views. Furthermore, we derive the upper bound on the estimation error of the proposed graph estimator and characterize the impact of the sample size, number of nodes, and the spectrum of the graph Laplacians on estimation errors. The approach is evaluated on synthetic graph data for robustness against noise, graph density, and sample size. Finally, the proposed framework is applied to two-view real-world graph data for graph learning and clustering.

## 1 Introduction

Many real-world data are represented through the relations between data samples, i.e., a graph structure (Newman (2018)). Although many datasets, including social networks and traffic networks, come with a known graph structure, there are a lot of applications where a graph is not readily available. For example, in many biological systems, e.g., gene regulatory networks (Li & Gui, 2006), the underlying graph structure is not directly observable. In such cases, inferring the topology of the graph is essential to analyze the data and model the relations.

Existing graph inference approaches are mostly limited to homogeneous datasets, where observed graph signals are assumed to be identically distributed and defined on a single graph. In many applications, the data may be heterogeneous or mixed and come from multiple related graphs, i.e., multiview graphs. In these situations, learning the topology of the views jointly by incorporating the relationships between views can improve performance (Tsai et al., 2022; Danaher et al., 2014; Navarro et al., 2022).

Traditional joint graphical structure inference methods are primarily based on Gaussian Graphical Models. These methods extend graphical lasso (Friedman et al. (2008)) to a joint learning setup, where they learn the precision matrices of multiple related Gaussian graphical models. They employ various penalties in the likelihood framework to exploit the common characteristics shared by different views (Guo et al. (2011); Danaher et al. (2014); Lee & Liu (2015); Mohan et al. (2014); Ma & Michailidis (2016); Huang & Chen (2015)). One prominent example of this approach is the joint graphical lasso (Danaher et al. (2014)), where fused or group lasso penalties are used to encourage topological similarity between views. However, these methods are limited by the assumption

that the observed graph signals are Gaussian, which is usually not true for real-world applications. Furthermore, they learn the precision matrices without imposing graph structure constraints on the learned views. These joint learning approaches have recently been extended to learn multiple graph Laplacian matrices instead of precision matrices (Yuan et al. (2023a); Zhang & Wang (2024a); Karaaslanli & Aviyente (2024)). However, all of these methods quantify the pairwise similarity between the views based on edge similarity. In many settings, such as gene regulatory networks (Mohan et al. (2014)), the differences between views may be better explained through the changes in the connectivity of a few nodes. This way of modeling the differences imposes a structure and provides an intuitive interpretation of the network differences.

In this paper, we introduce a joint graph Laplacian learning framework where the differences across the views are assumed to be driven by the perturbation to the individual nodes' connectivity across views. Based on this assumption, we introduce a Laplacian learning framework using the smoothness criterion, i.e., the graph signals are smooth or low frequency with respect to the underlying graph structure, with a regularization term that captures the node-based similarity across views. We focus on learning graphs for the case of two views where each view is assumed to be a perturbed version of the other by changing the connectivity of $r$ nodes with $r << n$. The corresponding optimization problem is solved using the Alternating Direction Method of Multipliers (ADMM). Finally, theoretical results are provided to quantify the upper bounds on the error between the estimated and true graph Laplacians as a function of the number of signals and nodes.

The main contributions of the proposed framework are:

- Extending structured multiview graph learning from GGM to smooth graph signals, where a valid graph topology instead of precision matrices is learned. Using smoothness, our framework is not restricted to GGMs and can handle different types of smooth graph signals.

- Introducing structure-based multiview graph learning, particularly a node perturbation model, in the context of smooth graph learning.

- Providing theoretical analysis and upper bounds on the estimation error of two-view Laplacian learning in terms of the graph size, sample size, and the radius around the true Laplacians, as the problem is non-convex. This estimation bound also suggests that the estimated values will not converge to the true values merely by increasing the sample size; rather, convergence also depends on the topology of the true graph structure.

## 2 BACKGROUND

### 2.1 RELATED WORK

Prior work in multiview graph learning has been mostly based on statistical models. These methods extend graphical lasso (Friedman et al. (2008)) to the joint learning case, where the precision matrices of multiple related GGMs are learned using various penalties in the likelihood framework to exploit the common characteristics shared by different views (Guo et al. (2011); Danaher et al. (2014); Lee & Liu (2015); Mohan et al. (2014); Ma & Michailidis (2016); Huang & Chen (2015)). The most notable among these is the joint graphical lasso (Danaher et al. (2014)), where fused or group lasso penalties are used to encourage topological similarity between views. However, these methods are limited by the assumption that the observed graph signals are Gaussian, suffer from increased computational complexity in the case of pairwise penalties and learn conditional dependencies instead of inferring the graph structure, which may not be suitable for subsequent learning tasks. Recently, these joint learning approaches have been extended to multiple graph Laplacian matrices (Yuan et al. (2023a)). However, this approach is still limited to Gaussian data and edge-based similarities across views.

The graph signal processing (GSP) community has recently addressed the problem of learning multiview graphs from heterogeneous data. This work can be divided into two categories, depending on whether one knows the association of the observed signals with the views *a priori*. In the first setup, multiple datasets are given and each dataset is defined in a view (Navarro et al. (2022); Navarro & Segarra (2022)). On the other hand, the second setup deals with the mixture of graph signals, where a single data set is given and the association of graph signals to the views is not known (Maretic

& Frossard (2020); Araghi et al. (2019); Karaaslanli & Aviyente (2022)). The focus of this paper is the first category. This problem setting has been studied most extensively to infer the topology of time-varying networks (Kalofolias et al. (2017); Yamada et al. (2019); Baingana & Giannakis (2016); Sardellitti et al. (2019)), where the aim is to learn graphs at multiple time points and to track changes in the structure of the graph over time. This problem can be posed as multiview graph learning with a regularization term that promotes pre-specified changes between consecutive graphs. More recently, the problem of multiview graph learning has been formulated with the assumption of graph stationarity (Navarro et al. (2022)). In this formulation, the signals are assumed to be stationary, and pairwise similarity between all graphs is used to regularize the optimization. In (Zhang & Wang (2024a)), the authors propose a multiview graph learning method based on the smoothness assumption. However, all of this prior work in GSP quantifies the similarity across views through edge-based similarity without considering the structure of the differences between views explicitly.

## 2.2 NOTATIONS

We represent a vector with bold lower case notation $\boldsymbol{x}$ and matrix of size $m \times n$ as $\mathbf{A} \in \mathbb{R}^{m \times n}$ with the $(i, j)$ th entry of the matrix $\mathbf{A}$ as $A_{ij} \ \forall \ i, j$. The trace of a square matrix $\mathbf{A} \in \mathbb{R}^{n \times n}$ is denoted as $\text{tr}(\mathbf{A}) = \sum_i A_{ii}$. The Frobenius norm $\| \cdot \|_F$ of a matrix $\mathbf{A}$, is defined as $\|\mathbf{A}\|_F = \sqrt{\sum_{i,j} A_{ij}^2}$ . $\| \cdot \|_{2,1}$ is the $\ell_{2,1}$ norm which is the the sum of $\ell_2$ norms of the rows of a matrix $\mathbf{A}$, i.e., $\|\mathbf{A}\|_{2,1} = \sum_i \sqrt{\sum_j A_{ij}^2}$. $\|\mathbf{A}\|_2$ is the spectral norm of matrix $\mathbf{A}$, that is the maximum sigular value of $\mathbf{A}$. The operator vec(.) is used for vectorization of matrix. The symbol $\odot$ is the Hadamard product (element-wise) product of two matrices and $\otimes$ is the Kronecker product of two matrices. $\mathcal{B}_r(\mathbf{A})$ is the open ball of radius $r$, with respect to the metric induced by Frobenius norm centered at the matrix $\mathbf{A}$. The all-one and all-zero vectors and matrices are denoted by $\mathbf{1}$ and $\mathbf{0}$, respectively. $|S|$ denotes the cardinality of a set $S$.

An undirected graph is defined as $G = (V, E)$, where $V$ is a set of $n$ nodes, i.e., $|V| = n$, and $E \subseteq V \times V$ is a set of edges. An edge connecting nodes $i$ and $j$ is represented as $E_{ij}$, with an associated weight $w_{ij}$. The graph $G$ can be algebraically represented using an $n \times n$ symmetric adjacency matrix $\mathbf{W} \in \mathbb{R}^{n \times n}$. Each element $W_{ij}$ is defined as $W_{ij} = W_{ji} = w_{ij}$ if $e_{ij} \in E$, and $W_{ij} = 0$ if there is no edge between nodes $i$ and $j$. The graph Laplacian is represented as $\mathbf{L} = \mathbf{D} - \mathbf{W}$, where $\mathbf{D}$ is the diagonal degree matrix, with each diagonal entry $D_{ii}$ calculated as $D_{ii} = \sum_{j=1}^n W_{ij}$. The eigendecomposition of $\mathbf{L}$ is given by $\mathbf{L} = \mathbf{U}^\top \mathbf{\Lambda} \mathbf{U}$, with $\mathbf{U}$ containing eigenvectors as columns and $\mathbf{\Lambda}$ is a diagonal matrix with diagonal elements $0 = \Lambda_{11} \leq \Lambda_{22} \leq \cdots \leq \Lambda_{nn}$.

## 3 SMOOTHNESS BASED GRAPH LEARNING

A graph signal defined on $G$ is a function $x : V \to \mathbb{R}$ and can be represented as a vector $\boldsymbol{x} \in \mathbb{R}^n$ where $x_i$ is the signal value on node $i$. Eigenvectors and eigenvalues of the Laplacian of $G$ can be used to define the graph Fourier transform (GFT), i.e., $\widehat{\boldsymbol{x}} = \mathbf{U}^\top \boldsymbol{x}$ where $\hat{x}_i$ is the Fourier coefficient at the $i$th frequency component $\Lambda_{ii}$. $\boldsymbol{x}$ is referred to as a smooth graph signal if most of the energy of $\widehat{\boldsymbol{x}}$ lies in low frequency components. The smoothness of $\boldsymbol{x}$ can then be quantified using the total variation of $\boldsymbol{x}$ measured in terms of the spectral density of its Fourier transform as:

$$\text{tr}(\widehat{\boldsymbol{x}}^\top \mathbf{\Lambda} \widehat{\boldsymbol{x}}) = \text{tr}(\boldsymbol{x}^\top \mathbf{U} \mathbf{\Lambda} \mathbf{U}^T \boldsymbol{x}) = \text{tr}(\boldsymbol{x}^\top \mathbf{L} \boldsymbol{x}). \tag{3.1}$$

## 3.1 SINGLE VIEW GRAPH LEARNING

An unknown graph $G$ can be learned from a set of graph signals defined on it based on some assumptions about the relation between the observed graph signals and the underlying graph structure. One such assumption is the smoothness of the observations with respect to $G$, which can be quantified using total variation equation 3.1. Total variation offers a natural criterion for finding the best topology in which observed signals have the desired smoothness property (Dong et al., 2019; Mateos et al., 2019).

Dong et. al. (Dong et al., 2016) proposed to learn $G$ by assuming the graph signals are smooth with respect to $G$. Given $\mathbf{X} \in \mathbb{R}^{n \times p}$ as the data matrix with the columns corresponding to the observed

graph signals, $G$ can be learned by minimizing smoothness with respect to the Laplacian matrix of $G$:

$$\underset{\mathbf{L}}{\text{minimize}} \ \text{tr}(\mathbf{X}^\top \mathbf{L} \mathbf{X}) + \alpha \|\mathbf{L}\|_F^2 \qquad \text{s.t.} \qquad \mathbf{L} \in \mathbb{L} \text{ and } \text{tr}(\mathbf{L}) = 2n, \qquad (3.2)$$

where the first term quantifies the total variation of graph signals and the second term controls the density of the learned graph such that larger values of hyperparameter $\alpha$ result in a denser graph. $\mathbf{L}$ is constrained to be in $\mathbb{L} = \{\mathbf{L} : \mathbf{L} \succeq \mathbf{0}, L_{ij} = L_{ji} \leq 0 \ \forall i \neq j, \mathbf{L1} = \mathbf{0}\}$, which is the set of valid Laplacians. The second constraint is added to prevent the trivial solution $\mathbf{L} = \mathbf{0}$.

# 4 PROBLEM FORMULATION: PERTURBED NODE MODEL FOR TWO-VIEW LEARNING (PN-TVL)

Given a set of signal samples for each view, $\mathbf{X}^{(k)} = [\boldsymbol{x}_1^{(k)}, \ldots, \boldsymbol{x}_{d_k}^{(k)}]$ where $\boldsymbol{x}_i^{(k)} \in \mathbb{R}^n$ and $\mathbf{X}^{(k)} \in \mathbb{R}^{n \times d_k}$, with $n$ being the number of nodes, $k \in \{1, 2\}$ the number of views and $d_k$ the number of signal samples in view $k$, the goal is to learn the individual graph structures, i.e., the graph Laplacians, $\mathbf{L}^{(k)}$. Assuming that the individual views differ due to particular nodes that are perturbed across the two views, thus have a completely different connectivity pattern, the problem of learning the individual graph Laplacians, $\mathbf{L}^{(k)}$, with the smoothness assumption can be expressed as

$$\min_{\mathbf{L}^{(k)}, \mathbf{V}} \sum_{k=1}^{2} \left[ \text{tr}({\mathbf{X}^{(k)}}^\top \mathbf{L}^{(k)} \mathbf{X}^{(k)}) + \gamma_1 \|\mathbf{L}^{(k)} - \mathbf{I} \odot \mathbf{L}^{(k)}\|_F^2 - \gamma_2 \text{tr}(\log(\mathbf{I} \odot \mathbf{L}^{(k)})) \right] + \gamma_3 \|\mathbf{V}\|_{2,1}$$

$$\text{s.t.} \quad \mathbf{L}^{(k)} \in \mathbb{L}, \quad \mathbf{L}^{(1)} - \mathbf{L}^{(2)} = \mathbf{V} + \mathbf{V}^\top,$$

$$(4.1)$$

where the first term quantifies the total variation of the observed signal, $\mathbf{X}^{(k)}$, with respect to the underlying graph Laplacian $\mathbf{L}^{(k)}$ similar to equation 3.2, the second term controls the sparsity of the learned graphs, the third term applies a logarithmic penalty to the degree of the learned graphs, $(\mathbf{I} \odot \mathbf{L}^{(k)})$, to ensure connectivity (Kalofolias et al., 2017) and the last term penalizes the difference between the two views, $\mathbf{V}$, using the row-column overlap norm (RCON) (Mohan et al., 2014). $\mathbf{L}^{(k)}$ is constrained to be in $\mathbb{L} = \{\mathbf{L} : \mathbf{L} \succeq \mathbf{0}, L_{ij} = L_{ji} \leq 0 \ \forall i \neq j, \mathbf{L1} = \mathbf{0}\}$, which is the set of valid Laplacians. In general, RCON applies $\ell_{q,p}$-norm to the difference of views, the primary objective being to identify nonzero rows and columns, with each row or column representing a perturbed node. In this work, we use $\ell_{2,1}$-norm such that the number of columns of $\mathbf{V}$ with non-zero $\ell_2$-norm corresponding to the perturbed nodes' connectivity is minimized.

The optimization problem in equation 4.1 is nonconvex due to the constraints. To deal with the nonconvex constraints, we present an equivalent form of the constraints $\mathbf{L}^{(k)} \succeq 0, \mathbf{L}^{(k)} \cdot \mathbf{1} = 0$ following (Zhao et al., 2019):

$$\mathbf{L}^{(k)} \succeq 0, \mathbf{L}^{(k)} \cdot \mathbf{1} = 0 \iff \mathbf{P}\mathcal{E}^{(k)}\mathbf{P}^\top, \mathcal{E}^{(k)} \succeq 0, \qquad (4.2)$$

where $\mathbf{P} \in \mathbb{R}^{n \times (n-1)}$ is the orthogonal complement of the vector $\mathbf{1}$, i.e., $\mathbf{P}^\top \mathbf{P} = \mathbf{I}$ and $\mathbf{P}^\top \mathbf{1} = \mathbf{0}$, and $\mathcal{E}^{(k)} \in \mathbb{R}^{(n-1) \times (n-1)}$ is a positive semi-definite matrix for the $k$th view. Note that the choice of $\mathbf{P}$ is nonunique. The equivalent objective function can be written as follows:

$$\min_{\mathcal{E}^{(k)}, \mathbf{V}} \sum_{k=1}^{2} \left[ \text{tr}({\mathbf{X}^{(k)}}^\top \mathbf{P}\mathcal{E}^{(k)}\mathbf{P}^\top \mathbf{X}^{(k)}) + \gamma_1 \|\mathbf{P}\mathcal{E}^{(k)}\mathbf{P}^\top - \mathbf{I} \odot \mathbf{P}\mathcal{E}^{(k)}\mathbf{P}^\top\|_F^2 - \gamma_2 \text{tr}(\log(\mathbf{I} \odot \mathbf{P}\mathcal{E}^{(k)}\mathbf{P}^\top)) \right]$$

$$+ \gamma_3 \|\mathbf{V}\|_{2,1}$$

$$\text{s.t.} \quad \mathbf{P}\mathcal{E}^{(1)}\mathbf{P}^\top - \mathbf{P}\mathcal{E}^{(2)}\mathbf{P}^\top = \mathbf{V} + \mathbf{V}^\top.$$

$$(4.3)$$

The optimization problem in equation 4.3 can then be solved using ADMM. The update steps for solving the PN-TVL optimization are given in Appendix A and the pseudocode is given in Algorithm 1.

---

**Algorithm 1** PN-TVL Optimization Algorithm

---

**Input:** $\gamma_1, \gamma_2, \gamma_3, \mu, \mathbf{P}, \mathbf{X}^{(k)}$.
**Output:** Laplacian matrices $\mathbf{L}^{(k)}$
 1: Initialize: Set $\mathcal{E}^{(k)}$, $\mathbf{V}$, and the auxiliary variables and dual variables to the zero matrix.
 2: **while** not converged **do**
 3:     Update $\mathcal{E}_{l+1}^{(k)}$ by equation A.4.
 4:     Update $\mathbf{Z}_{l+1}^{(k)}$ by equation A.6.
 5:     Update $\mathbf{C}_{l+1}^{(k)}$ by equation A.8.
 6:     Update $\mathbf{H}_{l+1}^{(1)}$ and $\mathbf{H}_{l+1}^{(2)}$ by equation A.10 and equation A.12 respectively.
 7:     Update $\mathbf{V}_{l+1}$ by equation A.14.
 8:     Update $\mathbf{Q}_{l+1}$ by equation A.16.
 9:     Update $\mathbf{W}_{l+1}$ by equation A.18.
10:     Update the Lagrange multipliers and penalty parameter by equation A.19.
11: **end while**

---

## 5 THEORETICAL ANALYSIS

To facilitate a unified analysis of multiple graph Laplacians, we define the parameter space as the set of block diagonal matrices in $\mathbb{R}^{3n \times 3n}$, where each diagonal block corresponds to the graph Laplacians $\mathbf{L}_1, \mathbf{L}_2$, and the matrix $\mathbf{V}$ following the same type of formulations of (Yuan et al., 2023b). For the simplicity of analysis, we consider $d_1 = d_2 = d$ throughout this section. To analyze the estimation error let us write down equation 4.1 after rescaling as follows,

$$
\sum_{k=1}^{2} \left[ \frac{1}{d} \operatorname{tr}(\mathbf{X}^{(k)^\top} \mathbf{L}^{(k)} \mathbf{X}^{(k)}) + \gamma_{1d} \|\mathbf{L}^{(k)} - \mathbf{I} \odot \mathbf{L}^{(k)}\|_F^2 - \gamma_{2d} \operatorname{tr}(\log(\mathbf{I} \odot \mathbf{L}^{(k)})) \right] + \gamma_{3d} \|\mathbf{V}\|_{2,1},
$$
(5.1)

subject to the constraints $\mathbf{L}^{(1)}, \mathbf{L}^{(2)} \in \mathbb{L} = \{\mathbf{L} \in \mathbb{R}^{n \times n} : \mathbf{L} \succeq 0, L_{ij} = L_{ji} \le 0, \mathbf{L} \cdot \mathbf{1} = 0\}$ and $\mathbf{L}^{(1)}, \mathbf{L}^{(2)}, \mathbb{C} \in \mathbb{C} = \{\mathbf{L}^{(1)}, \mathbf{L}^{(2)}, \mathbf{V} \in \mathbb{R}^{n \times n} : \mathbf{L}^{(1)} - \mathbf{L}^{(2)} = \mathbf{V} + \mathbf{V}^\top\}$. From Lemma B.7 we know that $\mathbb{L}$ is convex but the set of the constraints $\mathbb{C}$ is non-convex. Since the problem is inherently non-convex due to the constraint $\mathbf{L}_1 - \mathbf{L}_2 = \mathbf{V} + \mathbf{V}^\top$, we aim to derive estimation error bounds for local optima. To address this, we consider an additional restriction that the local minimizer $\widehat{\mathbf{L}}_\gamma \in \mathbb{R}^{3n \times 3n}$ is confined to a small neighborhood around the true solution $\mathbf{L}^* \in \mathbb{R}^{3n \times 3n}$. The block diagonals corresponding to $\widehat{\mathbf{L}}_\gamma$ and $\mathbf{L}^*$ are $\left(\widehat{\mathbf{L}}_\gamma^{(1)}, \widehat{\mathbf{L}}_\gamma^{(2)}, \widehat{\mathbf{V}}_\gamma\right)$ and $\left(\mathbf{L}^{(1^*)}, \mathbf{L}^{(2^*)}, \mathbf{V}^*\right)$.

We make the following assumptions to derive the theoretical results.

(A1) We assume that the optimization is constrained to a local neighborhood defined by an open ball of radius $r > 0$ centered at the true $3n \times 3n$ matrix $\mathbf{L}^*$ similar to the idea presented in (Loh, 2017). Mathematically, this can be expressed as the set:

$$
\mathcal{B}_r(\mathbf{L}^*) = \left\{ \mathbf{L} \in \mathbb{R}^{3n \times 3n} : \|\mathbf{L} - \mathbf{L}^*\|_F < r \right\}.
$$

Additionally we also assume that there exists $r^{(1)}, r^{(2)}, r^{(3)} > 0$, such that a constant

$$
K_r = \min \left\{ r^{(1)}, r^{(2)}, r^{(3)} : \left\| \mathbf{L}^{(k)} - \mathbf{L}^{(k)^*} \right\|_F < r^{(k)}, \|\mathbf{V} - \mathbf{V}^*\|_F < r^{(3)}, r^{(1)} + r^{(2)} + r^{(3)} < r \right\}.
$$
(5.2)

(A2) The set of signals $\left\{\mathbf{x}_j^{(k)}\right\}_{j=1}^{d}$ is assumed to follow a sub-Gaussian distribution (see B.1 for details) with mean $\mathbf{0}$ and covariance matrix $\Sigma^{(k)}$ for $k = 1, 2$. Using the sub-Gaussian assumption enhances the flexibility of the analysis, as it encompasses a broader range of distributions beyond the Gaussian case, while still retaining important concentration properties that make it suitable for statistical analysis.

(A3) Additionally, we assume that the maximum Frobenius norm of the true covariance matrices $\Sigma^{(1)^*}$ and $\Sigma^{(2)^*}$ is bounded by a positive constant $C_{\Sigma^*}$. Formally, this can be expressed as:

$$\max_{k \in \{1,2\}} \left\| \Sigma^{(k)^*} \right\|_F \leq C_{\Sigma^*}.$$

This constraint ensures that the true covariance matrices do not exhibit extreme values which could otherwise lead to numerical instability. To state the theorem let us first define the matrix $\mathbf{E}_{ii} \in \mathbb{R}^{n \times n}$ as an indicator matrix for the $i$-th diagonal element, defined as:

$$(\mathbf{E}_{ii})_{kl} = \begin{cases} 1, & \text{if } k = l = i, \\ 0, & \text{otherwise.} \end{cases}$$

**Theorem 5.1.** *Under the assumptions* $(A1), (A2), (A3)$ *with the regularization parameters* $\gamma_{1d}, \gamma_{2d}, \gamma_{3d} > 0$,

$$\left\| \widehat{\mathbf{L}}_\gamma - \mathbf{L}^* \right\|_F \leq \frac{4nr^2 C_r^2}{\tilde{\lambda}_{\gamma_d} \sqrt{d}} \max_{k \in \{1,2\}} C^{(k)} + \frac{2r^2 C_r^2}{\tilde{\lambda}_{\gamma_d}} \left\{ n^2 \gamma_{3d} + 2C_{\Sigma^*} + \tilde{\gamma}_d C_{\mathbf{L}} \right\}, \qquad (5.3)$$

*with probability at least* $1 - 2 \left\{ exp\left( -c^{(1)} (\sqrt{d} - 2c^{(1)}\sqrt{n})^2 \right) + exp\left( -c^{(2)}(\sqrt{d} - 2c^{(2)}\sqrt{n})^2 \right) \right\}$
*and* $d > 4n \max \left\{ c^{(1)^2}, c^{(2)^2} \right\}$ *where* $\tilde{\lambda}_{\gamma_d}$ *is the minimum eigenvalue of the matrix*

$$\mathbf{M} = \left[ \sum_{i=1}^{n} \gamma_{2d} \mathbf{L}_{ii}^{(k)^{*^{-2}}} \mathbf{E}_{ii} \otimes \mathbf{E}_{ii} + 2\gamma_{1d} \left( \mathbf{I}_n \otimes \mathbf{I}_n - \sum_{i=1}^{n} \mathbf{E}_{ii} \otimes \mathbf{E}_{ii} \right) \right],$$

$\tilde{\gamma}_d = 4\gamma_{1d} + 2\gamma_{2d}$, *the constant* $C_r$ *satisfies* $C_r \geq \frac{1}{2K_r}$ *and the constant* $C_{\mathbf{L}}$ *is dependent on the true graph structures as,*

$$C_{\mathbf{L}} = \max \left\{ \max_{k \in \{1,2\}} \left\| \mathbf{L}^{(k)^*} - \mathbf{I} \odot \mathbf{L}^{(k)^*} \right\|_F, \max_{k \in \{1,2\}} \left\| \left( \mathbf{I} \odot \mathbf{L}^{(k)^*} \right)^{-1} \right\|_F \right\},$$

*and* $c^{(1)}, c^{(2)}, C^{(1)}, C^{(2)} > 0$ *are the constants that depend on the sub-Gaussian norms* $\left\| \boldsymbol{x}^{(1)} \right\|_\psi$
*and* $\left\| \boldsymbol{x}^{(2)} \right\|_\psi$ *of a random vector taken from this distribution.*

The derived estimation error bound shows that the accuracy of the estimated $\widehat{\mathbf{L}}_\gamma$ is significantly influenced by the number of nodes $n$, the number of samples $d$, and the choice of regularization parameters. Similar to the results of Zhang & Wang (2024a) and Zhang & Wang (2024b), the upper bound has two parts where the first part is data dependent and the second part is completely dependent on the regularization parameters and on the topology of the true graph. Furthermore, the minimum eigenvalue $\tilde{\lambda}_{\gamma_d}$ of the matrix $\mathbf{M}$ plays a critical role in controlling the bound—small values can drastically increase the error, highlighting the importance of ensuring a well-conditioned matrix through appropriate selection of $\gamma_{1d}$, $\gamma_{2d}$, and $\gamma_{3d}$. Thus, balancing these factors is essential for achieving reliable estimation accuracy. The upper bound of the estimation error is also dependent on the radius $r$ around true $\mathbf{L}^*$ as there is a factor of $r^2$ is present in the numerator along with the constant $C_r$. Also, in this setup as the sets $\mathbb{L}$ and $\mathbb{C}$ have special properties, the projected unconstrained estimator will have the same estimation error bound as for the constrained estimator (For details see B.6 of the Appendix).

# 6 EXPERIMENTAL RESULTS

## 6.1 SIMULATED NETWORKS

We considered two random network models: Erdős-Rényi (ER) random network and random geometric graph (RGG). In ER graphs, node pairs are independently connected with probability $0.1$. For RGG, we used the setup from (Kalofolias, 2016), where 100 two-dimensional points are randomly drawn from $[0,1]^2$ and they are connected to each other with weight $\exp(-\|\boldsymbol{x}_i - \boldsymbol{x}_j\|_2^2/\sigma^2)$ where $\boldsymbol{x}_i$ is the coordinates of $i$th point and $\sigma = 0.25$. Weights smaller than $0.6$ are set to 0, while the remaining ones are set to 1 to generate binary graphs.

For each network model, we duplicated the adjacency matrix, $A$, into two matrices $A^{(1)}$ and $A^{(2)}$. We selected $r$ nodes at random for perturbation. For each selected node, we set the elements of the corresponding row and column of either $A^{(1)}$ or $A^{(2)}$ (chosen at random) to be i.i.d. drawn from a Bernoulli distribution. This results in $r$ perturbed nodes.

### 6.1.1 DATA GENERATION

Given the two view graphs, each $\mathbf{X}^k \in \mathbb{R}^{n \times d_k}$ is generated from $G^k$ using the smooth graph filter $h(\mathbf{L}^i)$ (Kalofolias, 2016). In particular, each column of $\mathbf{X}^k$ is generated as $\mathbf{X}^k_{:,j} = h(\mathbf{L}^k)\boldsymbol{x}_0$; where $\boldsymbol{x}_0 \sim \mathcal{N}(\mathbf{0}, \boldsymbol{I})$. In this paper, we considered three different types of graph filters: 1) Gaussian filter ($h(\mathbf{L}) = \mathbf{L}^\dagger$); 2) Heat filter ($h(\mathbf{L}) = \exp(-\alpha\mathbf{L})$, $\alpha = 5$ in this paper); and 3) Tikhonov filter ($h(\mathbf{L}) = (\boldsymbol{I} + \alpha\mathbf{L})^{-1}$, $\alpha = 20$ in this paper). In the case of the Gaussian filter, the resulting signals are Gaussian distributed and the graph Laplacian and the precision matrix are equivalent to each other. We finally add $\eta\%$ noise (in $\ell_2$ norm sense) to each generated $\mathbf{X}^i$. For each simulation, the average performance over 10 realizations is reported.

We evaluated the performance of our method for the two random graph models and three signal generation methods with respect to different simulation parameters. In particular, we evaluated the robustness of our method with respect to the number of signal samples ($d_k$), number of perturbed nodes ($r$) and noise level ($\eta$). In the first case, we fixed the number of nodes at $n = 100$, and the number of perturbed nodes at $r = 3$, and noise level $\eta = 0.1$, and varied the number of signal samples. For the second case, we fixed the number of nodes at $n = 100$, the number of samples at $d_k = 700$, and noise level $\eta = 0.1$, and varied the number of perturbed nodes. Finally, we fixed the number of nodes at $n = 100$, the number of samples at $d_k = 700$, and the number of perturbed nodes $r = 3$, and varied the noise level $\eta$.

### 6.1.2 BENCHMARK MODELS

We compare the proposed method with respect to the following methods:

- SV: Single view graph learning approach in (Dong et al., 2016) which learns the graph topology corresponding to each view independently by assuming that the signals are smooth with respect to each view's graph.
- MVGL: Multiview graph learning approach in (Karaaslanli & Aviyente, 2024) which jointly learns multiple graph Laplacians assuming the signals are smooth with respect to each view graph and the similarity between the views is enforced by minimizing the $\ell_1$-norm error between the view and the learned consensus graph.
- PNJGL: Perturbed node joint graph learning in (Mohan et al., 2014) that jointly learns precision matrices assuming a GGM where the similarity between the views is enforced by minimizing the row column overlap norm of the difference.

The performance of the methods is quantified by computing the average F1 score with respect to the ground truth graphs across runs.

## 6.2 RESULTS

Figure 1 displays the results for ER network under three different types of graph filter with varying numbers of samples, number of perturbed nodes, and the percentage of noise level. For all methods, the performance increases with increasing number of signal samples as expected with PN-TVL performing the best. While MVGL performs better than SV and PNJGL, its performance is lower than PN-TVL as the learning algorithm does not take the perturbation model into account. For small sample sizes, PNJGL performs better than SV illustrating the advantage of joint learning. However, as the sample size increases the improvement in PNJGL performance does not increase at the same rate with the other methods. This may be due to the difficulty of estimating larger size precision matrices. Similarly, the performance drops with a growing number of perturbed nodes and noise levels. While MVGL is robust against noise, it is not robust to increase in the number of perturbed nodes. This is because MVGL tries to maximize edge similarity. As the number of perturbed nodes increases, the number of dissimilar edges increases at a polynomial rate. Thus, the difference between the views becomes less sparse. PNJGL performs better than MVGL in these

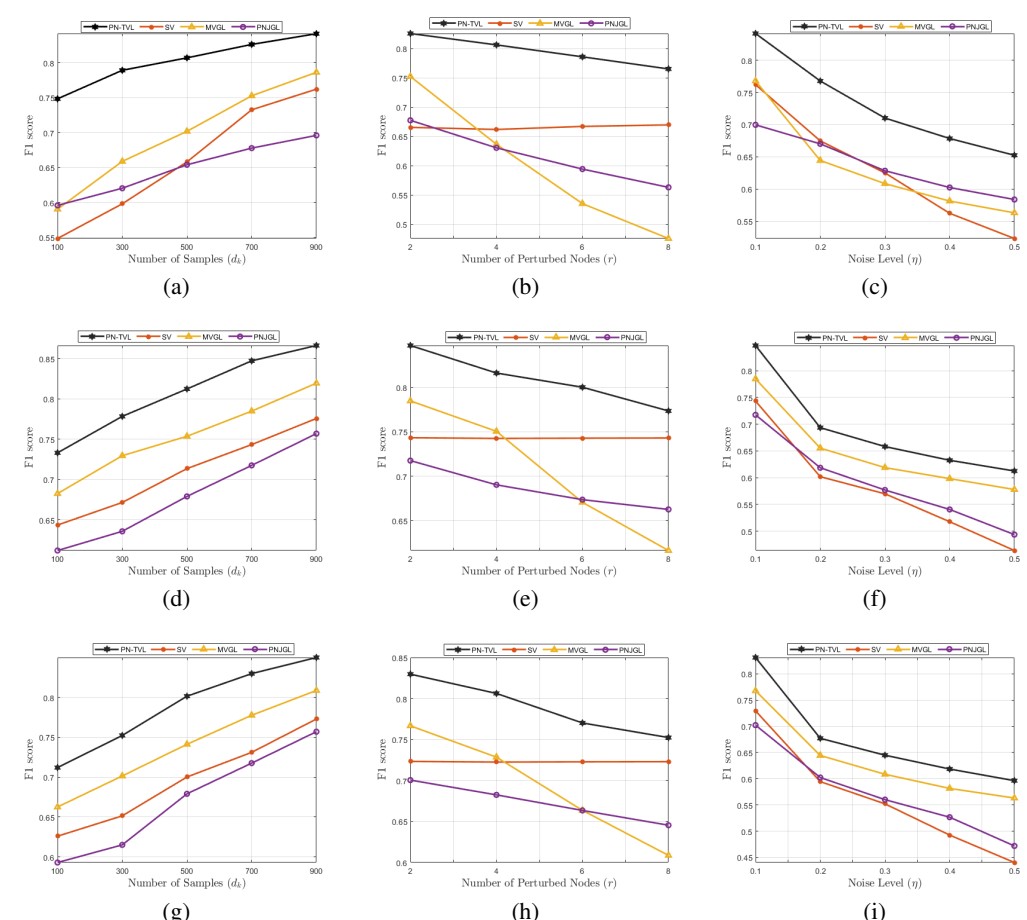

Figure 1: Comparison of performance for ER network model with three different graph filters for varying signal parameters: (a)-(c) varying number of samples $(d_k)$, number of perturbed nodes $(r)$, and noise level $(\eta)$, for Gaussian filter; (d)-(f) varying number of samples $(d_k)$, number of perturbed nodes $(r)$, and noise level $(\eta)$, for heat filter, (g)-(i) varying number of samples $(d_k)$, number of perturbed nodes $(r)$, and noise level $(\eta)$, for Tikhonov filter.

situations as it is based on the assumption that the view differences are driven by node connectivity. For the different graph filters, the performance of PN-TVL does not vary much showing that it is not sensitive to the distribution of the data but rather to the smoothness with respect to the graph. On the other hand, PNJGL is more robust to noise and performs better than MVGL and SV for Gaussian signal model (top row), while its performance is inferior to MVGL for non-Gaussian signals (middle and bottom rows in Figure 1). This is due to the fact PNJGL cannot accurately estimate precision matrices for non-Gaussian data.

The results for the RGGs are given in Appendix C.

### 6.2.1 SCALABILITY OF THE PROPOSED APPROACH

To evaluate the scalability of the proposed PN-TVL method, we generated a set of simulated networks with the number of nodes increasing from 50 to 800. The two view networks were generated using an ER model with $d_k = 700$, $r = 0.02n$ and $p = 0.1$. As shown in Figure 2, SV is the fastest method as it solves a simpler optimization problem with less added constraints. The run time for PN-TVL is comparable to the MVGL model and less than the PNJGL model since PNJGL uses Singular Value Decomposition (SVD) in every step of the solution. Despite the longer runtime compared to SV, PN-TVL provides the highest accuracy in graph learning.

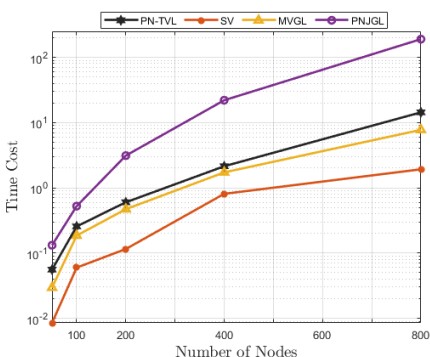

Figure 2: The scalability analysis of the PN-TVL model compared with other existing models as the number of nodes raised.

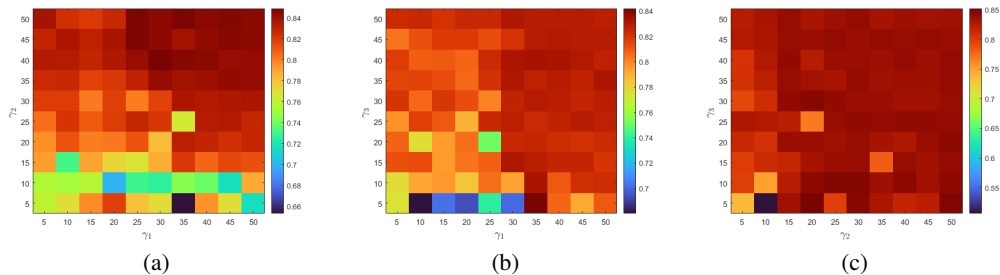

Figure 3: Sensitivity of the PN-TVL model in terms of NMI to the regularization parameters (a) vary $\gamma_1$, $\gamma_2$ and fix $\gamma_3$, (b) vary $\gamma_1$, $\gamma_3$ and fix $\gamma_2$, and (c) vary $\gamma_2$, $\gamma_3$ and fix $\gamma_1$.

### 6.2.2 SENSITIVITY TO HYPERPARAMETERS

In the proposed objective function 4.1, there are three parameters that are tuned, $(\gamma_1, \gamma_2, \gamma_3)$, where $\gamma_1$ controls the sparsity of the learned Laplacian matrices, $\gamma_2$ regulates the degree distribution (or connectivity) of the learned graphs, and $\gamma_3$ controls the norm of the difference between the views, i.e., the perturbation. In our implementation, we optimize the values of $(\gamma_1, \gamma_2, \gamma_3)$ using a grid search approach.

To assess the sensitivity of the PN-TVL model to these parameters, $\gamma_1$, $\gamma_2$, and $\gamma_3$ were varied from 5 to 50 in increments of 5, and the effect of adjusting two parameters at a time was analyzed while keeping the third fixed. As shown in Figure 3, the performance of the learning algorithm is most sensitive to the value of $\gamma_1$ which controls the sparsity of the learned graphs as when $\gamma_1$ is fixed, the performance does not change much for $\gamma_2$ and $\gamma_3$. This is expected since the sparsity of the learned graphs affect the accuracy of estimating $\mathbf{V}$ in addition to each view. Moreover, the optimal range for these parameters is found to be between 25 and 35.

### 6.3 REAL DATA ANALYSIS

In this section, we use a dataset of 2000 handwritten digit images from the UCI Machine Learning Repository (Dua et al., 2017) to demonstrate the performance of PN-TVL method in learning the graph structure and the cluster membership. The dataset includes samples from 10 distinct classes (digits $0-9$) with 6 features. We utilized two features: Fourier coefficients with dimension $d_1 = 76$ and Karhunen-Loève coefficients with dimension $d_2 = 64$, both derived from character shapes.

We learn each view graph from the corresponding features using PN-TVL and then apply spectral clustering to obtain the clustering labels. To assess the performance of the PN-TVL model, we use the commonly adopted evaluation metric, Normalized Mutual Information (NMI), and compare the results with other existing methods. Figure 4 shows average clustering accuracy across the two

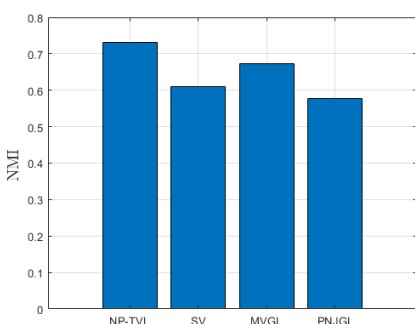

Figure 4: Average NMI results of spectral clustering applied to the graphs learned by different methods.

views for PN-TVL model compared to the other models. PN-TVL achieved an NMI of $0.73$ while the MVGL model has an NMI of $0.67$. This difference is because the MVGL model is designed to learn graphs based on edge-based similarity and may not learn the cluster structure well. In contrast, the SV model resulted in an average NMI of $0.61$ as it learns each view independently and does not consider the similarities between the views. Lastly, the PNJGL model has an average NMI of $0.57$, as it is designed to learn precision matrices rather than Laplacian matrices, so spectral clustering cannot accurately identify the clusters.

## 7 CONCLUSIONS

In this paper, we introduced a graph signal processing-based approach for learning graph Laplacians of two closely related graphs. The proposed approach is based on the assumptions that the observed graph signals are smooth with respect to the underlying graph structures, and that the differences between the two views are driven by the perturbation of a few nodes' connectivity. A cost function that enforces these two objectives is proposed and the corresponding optimization algorithm is presented. Theoretical results are presented on the upper bound of the estimation error for the inferred Laplacian as well as the view difference matrices. The proposed method is applied to both simulated and real graph-based data and compared to the state-of-the-art graph inference methods. The results show that PN-TVL is more robust regarding the increase in the number of perturbed nodes and added noise. Moreover, PN-TVL works well for various signal models, i.e., Gaussian and non-Gaussian, while graphical lasso-based methods like PNJGL's performance drops for non-Gaussian signals.

Future work will consider extending the proposed framework to multiple views. To keep computational complexity linear with respect to the number of views, we will develop consensus graph-based similarity constraints. Future work will also consider extending the proposed work to other node-based structural similarities, such as the co-hub model, where the views are assumed to have the same hub nodes.

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

# APPENDIX

## A  UPDATE STEPS FOR THE OPTIMIZATION ALGORITHM

In order to solve the optimization problem introduced in equation 4.1, we first introduce the auxiliary variables, $\mathbf{W}$, $\mathbf{H}^{(1)}$, $\mathbf{H}^{(2)}$, $\mathbf{Z}^{(k)}$, and $\mathbf{Q}$, to decouple the variables in the different terms. The optimization problem will be reformulated as follows:

$$
\min_{\substack{\mathbf{V},\mathbf{W},C^{(k)},\mathcal{E}^{(k)},\\ \mathbf{Z}^{(k)},\mathbf{Q},\mathbf{H}^{(k)}}} \sum_{k=1}^{2}\left[\mathrm{tr}(\mathbf{K}^{(k)}\mathcal{E}^{(k)}) + \gamma_1\|\mathbf{P}\mathcal{E}^{(k)}\mathbf{P}^\top - \mathbf{Z}^{(k)}\|_F^2 - \gamma_2\,\mathrm{tr}(\log(\mathbf{Z}^{(k)}))\right] + \gamma_3\|\mathbf{Q}\|_{2,1}
$$

$$
\text{s.t.}\quad \mathbf{C}^{(k)} = \mathbf{P}\mathcal{E}^{(k)}\mathbf{P}^\top, \mathbf{H}^{(1)} - \mathbf{H}^{(2)} = \mathbf{V} + \mathbf{W}, \mathbf{V} = \mathbf{W}^\top, \mathbf{H}^{(k)} = \mathbf{C}^{(k)}, \mathbf{V} = \mathbf{Q}, \mathbf{I}\odot\mathbf{C}^{(k)} = \mathbf{Z}^{(k)},
$$
(A.1)

where $\mathbf{K}^{(k)} = \mathbf{P}^\top\mathbf{X}^{(k)}{\mathbf{X}^{(k)}}^\top\mathbf{P}$. The augmented Lagrangian corresponding to equation A.1 can be written as follows:

$$
\min_{\substack{\mathbf{V},\mathbf{W},C^{(k)},\mathcal{E}^{(k)},\\ \mathbf{Z}^{(k)},\mathbf{Q},\mathbf{H}^{(k)}}} \sum_{k=1}^{2}\left[\mathrm{tr}(\mathbf{K}^{(k)}\mathcal{E}^{(k)}) + \gamma_1\|\mathbf{P}\mathcal{E}^{(k)}\mathbf{P}^\top - \mathbf{Z}^{(k)}\|_F^2 - \gamma_2\,\mathrm{tr}(\log(\mathbf{Z}^{(k)})) + \frac{\alpha}{2}\|\mathbf{C}^{(k)} - \mathbf{P}\mathcal{E}^{(k)}\mathbf{P}^\top + \frac{\mathbf{Y}^{(k)}}{\alpha}\|_F^2\right.
$$

$$
+ \frac{\alpha}{2}\|\mathbf{H}^{(k)} - \mathbf{C}^{(k)} + \frac{\mathbf{M}^{(k)}}{\alpha}\|_F^2 + \frac{\alpha}{2}\|\mathbf{Z}^{(k)} - \mathbf{I}\odot\mathbf{C}^{(k)} + \frac{\mathbf{U}^{(k)}}{\alpha}\|_F^2 + \gamma_3\|\mathbf{Q}\|_{2,1}
$$

$$
+ \frac{\alpha}{2}\|\mathbf{H}^{(1)} - \mathbf{H}^{(2)} - (\mathbf{V} + \mathbf{W}) + \frac{\mathbf{F}}{\alpha}\|_F^2 + \frac{\alpha}{2}\|\mathbf{V} - \mathbf{W}^\top + \frac{\mathbf{G}}{\alpha}\|_F^2 + \frac{\alpha}{2}\|\mathbf{V} - \mathbf{Q} + \frac{\mathbf{R}}{\alpha}\|_F^2,
$$
(A.2)

where $\mathbf{Y}^{(k)}$, $\mathbf{F}$, $\mathbf{G}$, $\mathbf{M}^{(k)}$, $\mathbf{R}$, $\mathbf{U}^{(k)}$ are the Lagrangian multipliers and $\alpha$ is the penalty parameter.

Equation A.2 can be solved by breaking it into multiple subproblems and optimizing each variable while keeping the others constant, as follows:

- **Subproblem $\mathcal{E}^{(k)}$:** To update $\mathcal{E}^{(k)}$, we fix all the other variables and consider the terms with $\mathcal{E}^{(k)}$ only as follows:

$$
\mathcal{E}_{l+1}^{(k)} = \min_{\mathcal{E}_l^{(k)}} \sum_{k=1}^{2}\left[\mathrm{tr}(\mathbf{K}^{(k)}\mathcal{E}_l^{(k)}) + \gamma_1\|\mathbf{P}\mathcal{E}_l^{(k)}\mathbf{P}^\top - \mathbf{Z}_l^{(k)}\|_F^2 + \frac{\alpha}{2}\|\mathbf{C}_l^{(k)} - \mathbf{P}\mathcal{E}_l^{(k)}\mathbf{P}^\top + \frac{\mathbf{Y}_l^{(k)}}{\alpha}\|_F^2\right].
$$
(A.3)

By taking the gradient with respect to $\mathcal{E}^{(k)}$, the solution of equation A.3 can be found as follows:

$$
\mathcal{E}_{l+1}^{(k)} = \frac{2\gamma_1\mathbf{P}^\top\mathbf{Z}_l^{(k)}P + \alpha\mathbf{P}^\top\mathbf{C}_l^{(k)}\mathbf{P} + \mathbf{P}^\top\mathbf{Y}_l^{(k)}\mathbf{P} - {\mathbf{K}^{(k)}}^\top}{2\gamma_1 + \alpha}.
$$
(A.4)

- **Subproblem $\mathbf{Z}^{(k)}$:** The solution of $\mathbf{Z}^{(k)}$ can be found by solving the following problem:

$$
\mathbf{Z}_{l+1}^{(k)} = \min_{\mathbf{Z}_l^{(k)}} \gamma_1\|\mathbf{Z}_l^{(k)} - \mathbf{P}\mathcal{E}_{l+1}^{(k)}\mathbf{P}^\top\|_F^2 - \gamma_2\,\mathrm{tr}(\log(\mathbf{Z}_l^{(k)})) + \frac{\alpha}{2}\|\mathbf{Z}_l^{(k)} - \mathbf{I}\odot\mathbf{C}_l^{(k)} + \frac{\mathbf{U}_l^{(k)}}{\alpha}\|_F^2.
$$
(A.5)

Similar to $\mathcal{E}^{(k)}$ subproblem, the solution of equation A.6 can be found as follows:

$$
\mathbf{Z}^{(k)} = \frac{\mathbf{B}_l^{(k)} + \sqrt{(\mathbf{B}_l^{(k)})^2 + 4(2\gamma_1 + \alpha)\gamma_2\mathbf{I}}}{2(2\gamma_1 + \alpha)},
$$
(A.6)

where $\mathbf{B}_l^{(k)} = 2\gamma_1\mathbf{P}\mathcal{E}_{l+1}^{(k)}\mathbf{P}^\top + \alpha\mathbf{I}\odot\mathbf{C}_l^{(k)} - \mathbf{U}_l^{(k)}$.

- **Subproblem $\mathbf{C}^{(k)}$**: The solution of $\mathbf{C}^{(k)}$ can be found by solving the following problem:

$$\mathbf{C}_{l+1}^{(k)} = \min_{\mathbf{C}_l^{(k)}} \frac{\alpha}{2} \|\mathbf{C}_l^{(k)} - \mathbf{P}\mathcal{E}_{l+1}^{(k)}\mathbf{P}^\top + \frac{\mathbf{Y}_l^{(k)}}{\alpha}\|_F^2 + \frac{\alpha}{2}\|\mathbf{C}_l^{(k)} - \mathbf{H}_l^{(k)} + \frac{\mathbf{M}_l^{(k)}}{\alpha}\|_F^2$$

$$+ \|\mathbf{I} \odot \mathbf{C}_l^{(k)} - \mathbf{Z}_l^{(k)} + \frac{\mathbf{U}_l^{(k)}}{\alpha}\|_F^2. \tag{A.7}$$

The problem in equation A.7 will update the diagonal and off-diagonal parts of the matrices, separately. To find the solution, the gradient with respect to $\mathbf{C}^{(k)}$ is taken and set to zero. The resulting solution can be expressed as follows:

$$\left[\mathbf{C}_{l+1}^{(k)}\right]_{ij} = \begin{cases} \frac{\alpha[\mathbf{P}\mathcal{E}^{(k)}\mathbf{P}^\top]_{ii} - Y_{ii}^{(k)} + \alpha H_{ii}^{(k)} + M_{ii}^{(k)} + Z_{ii}^{(k)} - U_{ii}^{(k)}}{3\alpha}, & \text{for} \quad i = j \\ \frac{\alpha[\mathbf{P}\mathcal{E}^{(k)}\mathbf{P}^\top]_{ij} - Y_{ij}^{(k)} + \alpha H_{ij}^{(k)} + M_{ij}^{(k)}}{2\alpha}, & \text{for} \quad i \neq j. \end{cases} \tag{A.8}$$

- **Subproblem $\mathbf{H}^{(1)}$ and $\mathbf{H}^{(2)}$**: In order to update $\mathbf{H}^{(1)}$, we fix all the other variables and consider the terms with $\mathbf{H}^{(1)}$ only as follows:

$$\mathbf{H}_{l+1}^{(1)} = \min_{\mathbf{H}_l^{(1)}} \frac{\alpha}{2}\|\mathbf{H}_l^{(1)} - \mathbf{H}_l^{(2)} - (\mathbf{V}_l + \mathbf{W}_l) + \frac{\mathbf{F}_l}{\alpha}\|_F^2 + \frac{\alpha}{2}\|\mathbf{H}_l^{(1)} - \mathbf{C}_{l+1}^{(1)} + \frac{\mathbf{M}_l^{(1)}}{\alpha}\|_F^2. \tag{A.9}$$

By taking the gradient of equation A.9 and setting it to zero, the solution of $\mathbf{H}^{(1)}$ can be written as follows:

$$\left[\mathbf{H}_{l+1}^{(1)}\right]_{ij} = \begin{cases} \left[\Gamma_h^{(1)} \odot \mathbf{I}\right]_+, & \text{for} \quad i = j \\ \left[\Gamma_h^{(1)} \odot (\mathbf{1}\mathbf{1}^\top - \mathbf{I})\right]_-, & \text{for} \quad i \neq j, \end{cases} \tag{A.10}$$

where $\Gamma_h^{(1)} = \frac{\alpha\mathbf{H}_l^{(2)} + \alpha\mathbf{V}_l + \alpha\mathbf{W}_l - \mathbf{F}_l + \alpha\mathbf{C}_{l+1}^{(1)} - \mathbf{M}_l^{(1)}}{2\alpha}$. The term $\left[\Gamma_h^{(1)} \odot \mathbf{I}\right]_+$ represents a diagonal matrix in which all elements are non-negative, where positive elements are retained and negative elements are set to zero. The term $\left[\Gamma_h^{(1)} \odot (\mathbf{1}\mathbf{1}^\top - \mathbf{I})\right]_-$ represents an off-diagonal matrix where only negative elements are kept, and positive elements are replaced by zeros. This ensures that $\mathbf{H}^{(1)}$ is a valid Laplacian matrix.

The solution of $\mathbf{H}^{(2)}$ can be found by solving the following problem:

$$\mathbf{H}_{l+1}^{(2)} = \min_{\mathbf{H}_l^{(2)}} \frac{\alpha}{2}\|\mathbf{H}_{l+1}^{(1)} - \mathbf{H}_l^{(2)} - (\mathbf{V}_l + \mathbf{W}_l) + \frac{\mathbf{F}_l}{\alpha}\|_F^2 + \frac{\alpha}{2}\|\mathbf{H}_l^{(2)} - \mathbf{C}_{l+1}^{(2)} + \frac{\mathbf{M}_l^{(2)}}{\alpha}\|_F^2. \tag{A.11}$$

Similar to $\mathbf{H}^{(1)}$, the solution of $\mathbf{H}^{(2)}$ can then be written as follows:

$$\left[\mathbf{H}_{l+1}^{(2)}\right]_{ij} = \begin{cases} \left[\Gamma_h^{(2)} \odot \mathbf{I}\right]_+, & \text{for} \quad i = j \\ \left[\Gamma_h^{(2)} \odot (\mathbf{1}\mathbf{1}^\top - \mathbf{I})\right]_-, & \text{for} \quad i \neq j, \end{cases} \tag{A.12}$$

where $\Gamma_h^{(2)} = \frac{\alpha\mathbf{H}^{(1)} - \alpha\mathbf{V}_l - \alpha\mathbf{W}_l + \mathbf{F}_l + \alpha\mathbf{C}_{l+1}^{(2)} - \mathbf{M}_l^{(2)}}{2\alpha}$.

- **Subproblem $\mathbf{V}$**: In order to update $\mathbf{V}$, we fix all the other variables and consider the terms with $\mathbf{V}$ only as follows:

$$\mathbf{V}_{l+1} = \min_{\mathbf{V}_l} \frac{\alpha}{2}\|\mathbf{H}_{l+1}^{(1)} - \mathbf{H}_{l+1}^{(2)} - (\mathbf{V}_l + \mathbf{W}_l) + \frac{\mathbf{F}_l}{\alpha}\|_F^2 + \frac{\alpha}{2}\|\mathbf{V}_l - \mathbf{Q}_l + \frac{\mathbf{R}_l}{\alpha}\|_F^2$$

$$+ \frac{\alpha}{2}\|\mathbf{V}_l - \mathbf{W}_l^\top + \frac{\mathbf{G}_l}{\alpha}\|_F^2. \tag{A.13}$$

By taking the derivative and setting it to zero, the solution of equation A.13 can be found as follows:

$$\mathbf{V}_{l+1} = \frac{\alpha\mathbf{H}_{l+1}^{(1)} - \alpha\mathbf{H}_{l+1}^{(2)} - \alpha\mathbf{W}_l + \mathbf{F}_l + \alpha\mathbf{Q}_l - \mathbf{R}_l + \alpha\mathbf{W}_l^\top - \mathbf{G}_l}{3\alpha}. \tag{A.14}$$

- **Subproblem Q**: In order to update $\mathbf{Q}$, we fix all the other variables and consider the terms with $\mathbf{Q}$ only as follows:

$$\mathbf{Q}_{l+1} = \min_{\mathbf{Q}_l} \gamma_3\|\mathbf{Q}_l\|_{2,1} + \frac{\alpha}{2}\|\mathbf{Q}_l - \mathbf{V}_{l+1} + \frac{\mathbf{R}_l}{\alpha}\|_F^2. \tag{A.15}$$

The solution of equation A.15 can be found by using the proximal operator for $\ell_{2,1}$-norm as follows (Liu et al., 2012):

$$\mathbf{Q}_{l+1} = \mathcal{T}_{2,1}(\mathbf{V}_{l+1} + \frac{\mathbf{R}_l}{\alpha}, \frac{\gamma_3}{\alpha}). \tag{A.16}$$

- **Subproblem W**: The solution of $\mathbf{W}_{l+1}$ can be found by solving the follows minimization problem:

$$\mathbf{W}_{l+1} = \min_{\mathbf{W}_l} \frac{\alpha}{2}\|\mathbf{V}_{l+1} + \mathbf{W}_{l+1} - (\mathbf{H}_{l+1}^{(1)} - \mathbf{H}_{l+1}^{(2)}) + \frac{\mathbf{F}_l}{\alpha}\|_F^2 + \frac{\alpha}{2}\|\mathbf{W}_{l+1}^\top - \mathbf{V}_{l+1} + \frac{\mathbf{G}_l}{\alpha}\|_F^2. \tag{A.17}$$

Similar to the $\mathbf{V}$ subproblem, the solution of equation A.17 can be found as follows:

$$\mathbf{W}_{l+1} = \frac{\alpha\mathbf{H}_{l+1}^{(1)} - \alpha\mathbf{H}_{l+1}^{(2)} - \alpha\mathbf{V}_{l+1} - \mathbf{F}_l + \alpha\mathbf{V}_{l+1}^\top - \mathbf{G}_l^\top}{2\alpha}. \tag{A.18}$$

Finally, the Lagrangian multipliers and the penalty parameters can be updated as follows:

$$\begin{aligned}
\mathbf{Y}_{l+1}^{(k)} &= \mathbf{Y}_l^{(k)} + \alpha_l(\mathbf{C}_{l+1}^{(k)} - \mathbf{P}\mathcal{E}_{l+1}^{(k)}\mathbf{P}^\top), \\
\mathbf{U}_{l+1}^{(k)} &= \mathbf{U}_l^{(k)} + \alpha_l(\mathbf{Z}_{l+1}^{(k)} - \mathbf{I}\odot\mathbf{C}_{l+1}^{(k)}), \\
\mathbf{M}_{l+1}^{(k)} &= \mathbf{M}_l^{(k)} + \alpha_l(\mathbf{H}_{l+1}^{(k)} - \mathbf{C}_{l+1}^{(k)}), \\
\mathbf{F}_{l+1} &= \mathbf{F}_l + \alpha_l(\mathbf{H}_{l+1}^{(1)} - \mathbf{H}_{l+1}^{(2)} - (\mathbf{V}_{l+1} + \mathbf{W}_{l+1})), \\
\mathbf{G}_{l+1} &= \mathbf{G}_l + \alpha_l(\mathbf{V}_{l+1} - \mathbf{W}_{l+1}^\top), \\
\mathbf{R}_{l+1} &= \mathbf{R}_l + \alpha_l(\mathbf{V}_{l+1} - \mathbf{Q}_{l+1}), \\
\alpha_{l+1} &= \mu\alpha_l, \mu > 1.
\end{aligned} \tag{A.19}$$

# B  APPENDIX PART FOR THEORETICAL ANALYSIS

## B.1  SOME DEFINITIONS

**Sub-Gaussian Random Vector:** A random vector $\boldsymbol{x} \in \mathbb{R}^n$ is called **sub-Gaussian** if all of its one-dimensional marginals are sub-Gaussian. Specifically, $\boldsymbol{x}$ is sub-Gaussian if there exists a constant $K > 0$ such that, for any unit vector $\mathbf{u} \in \mathbb{R}^n$,

$$\mathbb{E}\left[e^{t\mathbf{u}^\top(\boldsymbol{x}-\mathbb{E}[\boldsymbol{x}])}\right] \leq e^{\frac{K^2 t^2}{2}}, \quad \forall t \in \mathbb{R}.$$

**Sub-Gaussian Norm of a Random Vector:** The sub-Gaussian norm of a random vector $\boldsymbol{x} \in \mathbb{R}^n$, denoted as $\|\boldsymbol{x}\|_{\psi_2}$, is defined as:

$$\|\boldsymbol{x}\|_{\psi_2} = \sup_{\mathbf{u}\in\mathbb{S}^{n-1}} \|\mathbf{u}^\top\boldsymbol{x}\|_{\psi_2},$$

where $\mathbb{S}^{n-1}$ is the unit sphere in $\mathbb{R}^p$.

## B.2 SOME IMPORTANT RESULTS ON MATRICES

**Lemma B.1.** *The matrix $\mathbf{M} \in \mathbb{R}^{n^2 \times n^2}$ is positive definite if $\gamma_{1d} > 0$ and $\gamma_{2d} > 0$ when $\mathbf{M}$ is as the following.*

$$\mathbf{M} = \sum_{i=1}^{n} \gamma_{2d} \mathbf{L}_{ii}^{(k)^{*^{-2}}} \mathbf{E}_{ii} \otimes \mathbf{E}_{ii} + 2\gamma_{1d} \left( \mathbf{I}_n \otimes \mathbf{I}_n - \sum_{i=1}^{n} \mathbf{E}_{ii} \otimes \mathbf{E}_{ii} \right). \tag{B.1}$$

*Proof.* For any non-zero vector $\mathbf{y} \in \mathbb{R}^{n^2}$

$$\mathbf{y}^\top \mathbf{M} \mathbf{y} = \sum_i \gamma_{2d} \mathbf{L}_{ii}^{(k)^{*^{-2}}} y_{ii}^2 + 2\gamma_{1d} \sum_{i \neq j} y_{ij}^2.$$

Since both $\gamma_{2d} > 0$ and $\gamma_{1d} > 0$, the quadratic form is strictly positive for any non-zero vector $\mathbf{y}$. So, $\mathbf{M}$ is positive definite matrix.

$\square$

**Lemma B.2.** *For any two symmetric matrices $\mathbf{A}, \mathbf{B} \in \mathbb{R}^{n \times n}$,*

$$\mathrm{tr}(\mathbf{A}\mathbf{B}) \leq \|\mathbf{A}\|_F \|\mathbf{B}\|_F.$$

*Proof.* Since $\mathbf{A}$ and $\mathbf{B}$ are symmetric, $\mathbf{A}^T = \mathbf{A}$ and $\mathbf{B}^T = \mathbf{B}$. Thus,

$$\langle \mathbf{A}, \mathbf{B} \rangle = \mathrm{tr}(\mathbf{A}\mathbf{B}).$$

Using Cauchy-Schwarz inequality and the sub-multiplicative property of Frobenius norm, we have,

$$\langle \mathbf{A}, \mathbf{B} \rangle \leq \|\mathbf{A}\mathbf{B}\|_F \leq \|\mathbf{A}\|_F \|\mathbf{B}\|_F.$$

Hence, the proof. $\square$

**Lemma B.3.** *Suppose $\mathbf{A}, \mathbf{B} \in \mathbb{R}^{n \times n}$ then for the $\ell_{2,1}$ norm of the product of matrices, the following inequality holds.*

$$\|\mathbf{A}\mathbf{B}\|_{2,1} \leq n \|\mathbf{A}\|_F \|\mathbf{B}\|_1.$$

*Proof.* We can write the $l_{2,1}$ norm of $\mathbf{A}\mathbf{B}$ matrix by expanding it as,

$$\|\mathbf{A}\mathbf{B}\|_{2,1} = \sum_{i=1}^{n} \left( \sum_{j=1}^{n} \left( \sum_{k=1}^{n} a_{ik} b_{kj} \right)^2 \right)^{1/2}$$

$$\leq \sum_{i=1}^{n} \sum_{j=1}^{n} \left| \sum_{k=1}^{n} a_{ik} b_{kj} \right|$$

$$= \|\mathbf{A}\mathbf{B}\|_1 \leq \|\mathbf{A}\|_1 \|\mathbf{B}\|_1 \leq n \|\mathbf{A}\|_F \|\mathbf{B}\|_1. \tag{B.2}$$

$\square$

**Lemma B.4.** *For a square matrix $\mathbf{A} \in \mathbb{R}^{n \times n}$,*

$$\|\mathbf{A}\|_F \leq \sqrt{n} \cdot \|\mathbf{A}\|_2. \tag{B.3}$$

*Proof.* Let $\sigma_1, \sigma_2, \ldots, \sigma_n$ be the singular values of the matrix $\mathbf{A}$. Then, the Frobenius norm can be expressed as,

$$\|\mathbf{A}\|_F = \sqrt{\sum_{i=1}^{n} \sigma_i^2}.$$

Similarly, the operator norm (spectral norm) is given by the largest singular value:

$$\|\mathbf{A}\|_2 = \sigma_1 \quad (\text{assuming } \sigma_1 \geq \sigma_2 \geq \cdots \geq \sigma_n).$$

The rest of the proof is as follows,

$$\|\mathbf{A}\|_F = \sqrt{\sum_{i=1}^{n} \sigma_i^2} \leq \sqrt{n \cdot \sigma_1^2} = \sqrt{n}\sigma_1 = \sqrt{n}\|\mathbf{A}\|_2. \tag{B.4}$$

$\square$

**Lemma B.5.** *Given two symmetric positive semi-definite matrices $\mathbf{A}$ and $\mathbf{B}$ of order $n \times n$ we have,*

$$\mathrm{tr}(\mathbf{AB}) \geq \lambda_{\min}(\mathbf{A}) \cdot \mathrm{tr}(\mathbf{B}).$$

*where $\lambda_{\min}(\mathbf{A})$ is the minimum eigenvalue of the matrix $\mathbf{A}$*

*Proof.* Let $\mathbf{A}$ be a symmetric matrix with eigenvalues $\lambda_1, \lambda_2, \ldots, \lambda_n$, and let $\lambda_{\min}(\mathbf{A})$ denote the smallest eigenvalue of $\mathbf{A}$. Now, since $\mathbf{A}$ and $\mathbf{B}$ both are positive semi-definite and symmetric we have,

$$\mathrm{tr}(\mathbf{AB}) = \mathrm{tr}(\sqrt{\mathbf{A}}\sqrt{\mathbf{A}^\top}\mathbf{B}) = \mathrm{tr}(\sqrt{\mathbf{A}}\mathbf{B}\sqrt{\mathbf{A}^\top}) = \sum_{i=1}^{n} \mathbf{e}_i^\top \sqrt{\mathbf{A}}\mathbf{B}\sqrt{\mathbf{A}^\top}\mathbf{e}_i \leq 0. \tag{B.5}$$

where $\mathbf{e}_i$ is the n-dimensional vector with i-th element equal to 1 and other elements are zero. Now, we can decompose $\mathbf{A}$ as follows,

$$\mathbf{A} = \lambda_{\min}(\mathbf{A})\mathbf{I}_n + (\mathbf{A} - \lambda_{\min}(\mathbf{A})\mathbf{I}_n).$$

So, using B.5 we have the following,

$$\mathrm{tr}(\mathbf{AB}) = \lambda_{\min}(\mathbf{A})\,\mathrm{tr}(\mathbf{B}) + \mathrm{tr}((\mathbf{A} - \lambda_{\min}(\mathbf{A})\mathbf{I}_n)\mathbf{B}) \geq \lambda_{\min}(\mathbf{A})\,\mathrm{tr}(\mathbf{B}). \tag{B.6}$$

as $(\mathbf{A} - \lambda_{\min}(\mathbf{A})\mathbf{I}_n)$ is a positive semi definite matrix. $\square$

## B.3 An Important Result on the Error Bound of Sample Covariance and the True Covariance

**Lemma B.6.** *Assume that $\mathbf{A}$ is an $N \times n$ matrix whose rows $\mathbf{A}_i$ are independent sub-gaussian random vectors in $\mathbb{R}^n$ with second moment matrix $\Sigma$. Then for every $t \geq 0$, the following inequality holds with probability at least $1 - 2\exp(-ct^2)$:*

$$\left\| \frac{1}{N}\mathbf{A}^\top\mathbf{A} - \Sigma \right\|_2 \leq \max(\delta, \delta^2) \quad where \quad \delta = C\sqrt{\frac{n}{N}} + \frac{t}{\sqrt{N}}.$$

*Here, as before, $C = C_K$, $c = c_K > 0$ depend only on the sub-gaussian norm $K = \max_i \|\mathbf{A}_i\|_{\psi_2}$ of the rows.*

*Proof.* For the proof of this Lemma on the estimation error bound of sample covariance matrix from the true covariance matrix, see the theorem 5.39 of (Vershynin, 2011). $\square$

## B.4 Analysis of the Sets of Constraints

**Lemma B.7.**

$$\mathbb{L} = \left\{ \mathbf{L} \in \mathbb{R}^{n \times n} : \mathbf{L} \succeq \mathbf{0},\ L_{ij} = L_{ji} \leq 0\ \forall\, i \neq j,\ \mathbf{L} \cdot \mathbf{1} = \mathbf{0} \right\}$$

*is **convex** and **closed** set.*

*Proof.* CONVEXITY OF THE SET $\mathbb{L}$

A set $\mathcal{S}$ is convex if for any two matrices $\mathbf{L}_1, \mathbf{L}_2 \in \mathcal{S}$ and any scalar $\lambda \in [0, 1]$, the convex combination $\lambda \mathbf{L}_1 + (1 - \lambda)\mathbf{L}_2$ is also in $\mathcal{S}$. That is, for any $\mathbf{L}_1, \mathbf{L}_2 \in \mathbb{L}$, we need to check whether $\lambda \mathbf{L}_1 + (1 - \lambda)\mathbf{L}_2 \in \mathbb{L}$.

At first, to prove that the set of positive semi-definite matrices is convex that is if $\mathbf{L}_1 \succeq 0$ and $\mathbf{L}_2 \succeq 0$, then for any $\lambda \in [0, 1]$,
$$\lambda \mathbf{L}_1 + (1 - \lambda)\mathbf{L}_2 \succeq \mathbf{0}.$$
Therefore, the positive semi-definiteness constraint is preserved under convex combinations. For the constraint $L_{ij} = L_{ji} \leq 0 \; \forall \, i \neq j$ is convex. This is because if $L_{ij}^{(1)} \leq 0$ and $L_{ij}^{(2)} \leq 0$ for all $i \neq j$, then for any $\lambda \in [0, 1]$,
$$L_{ij} = \lambda L_{ij}^{(1)} + (1 - \lambda)L_{ij}^{(2)} \leq 0.$$
Hence, this constraint is preserved under convex combinations. The constraint $\mathbf{L} \cdot \mathbf{1} = 0$ (i.e., each row sums to zero) is linear, and the set of matrices whose rows sum to zero is an affine subspace, which is convex. That is, if $\mathbf{L}_1 \cdot \mathbf{1} = \mathbf{0}$ and $\mathbf{L}_2 \cdot \mathbf{1} = \mathbf{0}$, then for any $\lambda \in [0, 1]$,
$$(\lambda \mathbf{L}_1 + (1 - \lambda)\mathbf{L}_2) \cdot \mathbf{1} = \lambda(\mathbf{L}_1 \cdot \mathbf{1}) + (1 - \lambda)(\mathbf{L}_2 \cdot \mathbf{1}) = \mathbf{0}.$$
Thus, the zero row sum constraint is also preserved under convex combinations. Since each of the individual constraints is convex, the intersection of these constraints is also convex. Therefore, the set $\mathbb{L}$ is **convex**.

CLOSEDNESS OF THE SET $\mathbb{L}$

To show that $\mathbb{L}$ is **closed**, we need to check whether it contains all its limit points. That is, if we have a sequence of matrices $\{\mathbf{L}_m\} \subset \mathbb{L}$ such that $\mathbf{L}_m \to \mathbf{L}$ (in some matrix norm, such as the Frobenius norm), we need to check whether $\mathbf{L} \in \mathbb{L}$. If $\mathbf{L}_m \succeq \mathbf{0}$ and $\mathbf{L}_n \to \mathbf{L}$, then $\mathbf{L} \succeq \mathbf{0}$. This is because the eigenvalues of $\mathbf{L}_n$ (which are non-negative) converge to the eigenvalues of $\mathbf{L}$, ensuring that $\mathbf{L} \succeq \mathbf{0}$. If $L_{ij}^{(m)} = L_{ji}^{(m)} \leq 0$ for all $i \neq j$, and $\mathbf{L}_m \to \mathbf{L}$, then by continuity of the matrix entries, we have $L_{ij} = L_{ji} \leq 0 \; \forall \, i \neq j$. If $\mathbf{L}_m \cdot \mathbf{1} = \mathbf{0}$ for all $m$, and $\mathbf{L}_m \to \mathbf{L}$, then by the continuity of matrix-vector multiplication, $\mathbf{L} \cdot \mathbf{1} = \mathbf{0}$. Since each of the individual constraints defines a closed set, and the intersection of closed sets is closed, the set $\mathbb{L}$ is **closed**. $\square$

**Lemma B.8.**
$$\mathbb{C} = \left\{ \mathbf{L}^{(1)}, \mathbf{L}^{(2)}, \mathbf{V} \in \mathbb{R}^{n \times n} : \mathbf{L}^{(1)} - \mathbf{L}^{(2)} = \mathbf{V} + \mathbf{V}^{\top} \right\}$$
*is closed.*

*Proof.* We want to show that the set
$$\mathbb{C} = \left\{ \mathbf{L}^{(1)}, \mathbf{L}^{(2)}, \mathbf{V} \in \mathbb{R}^{n \times n} : \mathbf{L}^{(1)} - \mathbf{L}^{(2)} = \mathbf{V} + \mathbf{V}^{\top} \right\}$$
is closed. The set $\mathbb{C}$ consists of triplets $(\mathbf{L}^{(1)}, \mathbf{L}^{(2)}, \mathbf{V}) \in \mathbb{R}^{n \times n}$ that satisfy the constraint:
$$\mathbf{L}^{(1)} - \mathbf{L}^{(2)} = \mathbf{V} + \mathbf{V}^{\top}.$$
This is a linear equality constraint that must hold between $\mathbf{L}^{(1)}$, $\mathbf{L}^{(2)}$, and $\mathbf{V}$. Suppose we have a sequence $\{(\mathbf{L}_m^{(1)}, \mathbf{L}_m^{(2)}, \mathbf{V}_m)\} \subset \mathbb{C}$, which converges to a limit $(\mathbf{L}^{(1)}, \mathbf{L}^{(2)}, \mathbf{V})$. That is, as $m \to \infty$,
$$\mathbf{L}_m^{(1)} \to \mathbf{L}^{(1)}, \quad \mathbf{L}_m^{(2)} \to \mathbf{L}^{(2)}, \quad \mathbf{V}_m \to \mathbf{V}$$
in some norm (e.g., the Frobenius norm). Since $(\mathbf{L}_m^{(1)}, \mathbf{L}_m^{(2)}, \mathbf{V}_m) \in \mathbb{C}$, for each $m$, the constraint $\mathbf{L}_m^{(1)} - \mathbf{L}_m^{(2)} = \mathbf{V}_m + \mathbf{V}_m^{\top}$ holds. That is:
$$\mathbf{L}_m^{(1)} - \mathbf{L}_m^{(2)} = \mathbf{V}_m + \mathbf{V}_m^{\top} \quad \forall m.$$
Now, taking the limit as $m \to \infty$, and using the continuity of matrix addition and transposition, we get:
$$\lim_{m \to \infty} (\mathbf{L}_m^{(1)} - \mathbf{L}_m^{(2)}) = \lim_{m \to \infty} (\mathbf{V}_m + \mathbf{V}_m^{\top}).$$

Since both sides of the equality converge (because the sequence converges), we can exchange the limit with the operations of addition and transposition:

$$\mathbf{L}^{(1)} - \mathbf{L}^{(2)} = \mathbf{V} + \mathbf{V}^\top.$$

The limit point $(\mathbf{L}^{(1)}, \mathbf{L}^{(2)}, \mathbf{V})$ satisfies the same constraint as the elements of the sequence, which means that the limit point belongs to the set $\mathbb{C}$. Therefore, the set $\mathbb{C}$ contains all its limit points, and hence it is **closed**.

$\square$

### B.5 PROOF OF THE THEOREM 5.1

*Proof.* Since $\widehat{\mathbf{L}}_\gamma$ is a local minimizer around the small neighborhood of $\mathbf{L}^*$, then for any $\mathbf{L} \in \mathcal{B}_r(\mathbf{L}^*)$,

$$\sum_{k=1}^2 \left[ \frac{1}{d} \operatorname{tr}(\mathbf{X}^{(k)\top} \widehat{\mathbf{L}}_\gamma^{(k)} \mathbf{X}^{(k)}) + \gamma_{1d} \|\widehat{\mathbf{L}}_\gamma^{(k)} - \mathbf{I} \odot \widehat{\mathbf{L}}_\gamma^{(k)}\|_F^2 - \gamma_{2d} \operatorname{tr}(\log(\mathbf{I} \odot \widehat{\mathbf{L}}_\gamma^{(k)})) \right] + \gamma_{3d} \|\widehat{\mathbf{V}}_\gamma\|_{2,1}$$

$$\leq \sum_{k=1}^2 \left[ \frac{1}{d} \operatorname{tr}(\mathbf{X}^{(k)\top} \mathbf{L}^{(k)} \mathbf{X}^{(k)}) + \gamma_{1d} \|\mathbf{L}^{(k)} - \mathbf{I} \odot \mathbf{L}^{(k)}\|_F^2 - \gamma_{2d} \operatorname{tr}(\log(\mathbf{I} \odot \mathbf{L}^{(k)})) \right] + \gamma_{3d} \|\mathbf{V}\|_{2,1} . \tag{B.7}$$

We define $\widehat{\boldsymbol{\Sigma}}^{(k)} = \frac{1}{d} \mathbf{X}^{(k)} \mathbf{X}^{(k)\top}$, and then rewrite the above inequality as,

$$\sum_{k=1}^2 \operatorname{tr}((\widehat{\mathbf{L}}_\gamma^{(k)} - \mathbf{L}^{(k)}) \widehat{\boldsymbol{\Sigma}}^{(k)}) + \gamma_{1d} \|\widehat{\mathbf{L}}_\gamma^{(k)} - \mathbf{I} \odot \widehat{\mathbf{L}}_\gamma^{(k)}\|_F^2 - \gamma_{2d} \operatorname{tr}(\log(\mathbf{I} \odot \widehat{\mathbf{L}}_\gamma^{(k)})) - \gamma_{1d} \|\mathbf{L}^{(k)} - \mathbf{I} \odot \mathbf{L}^{(k)}\|_F^2$$

$$+ \gamma_{2d} \operatorname{tr}(\log(\mathbf{I} \odot \mathbf{L}^{(k)})) \leq \gamma_{3d} \left\{ \|\mathbf{V}\|_{2,1} - \|\widehat{\mathbf{V}}_\gamma\|_{2,1} \right\} .$$

If $\boldsymbol{\Sigma}^{(k)}$ is the covariance matrix corresponding to $\mathbf{L}^{(k)}$ then, we further have,

$$\sum_{k=1}^2 \operatorname{tr}((\widehat{\mathbf{L}}_\gamma^{(k)} - \mathbf{L}^{(k)}) \boldsymbol{\Sigma}^{(k)}) + \gamma_{1d} \|\widehat{\mathbf{L}}_\gamma^{(k)} - \mathbf{I} \odot \widehat{\mathbf{L}}_\gamma^{(k)}\|_F^2 - \gamma_{2d} \operatorname{tr}(\log(\mathbf{I} \odot \widehat{\mathbf{L}}_\gamma^{(k)})) - \gamma_{1d} \|\mathbf{L}^{(k)} - \mathbf{I} \odot \mathbf{L}^{(k)}\|_F^2$$

$$+ \gamma_{2d} \operatorname{tr}\left(\log\left(\mathbf{I} \odot \mathbf{L}^{(k)}\right)\right) \leq \gamma_{3d} \left(\|\mathbf{V}\|_{2,1} - \|\widehat{\mathbf{V}}_\gamma\|_{2,1}\right) + \sum_{k=1}^2 \operatorname{tr}\left((\mathbf{L}^{(k)} - \widehat{\mathbf{L}}^{(k)})\left(\widehat{\boldsymbol{\Sigma}}^{(k)} - \boldsymbol{\Sigma}^{(k)}\right)\right) . \tag{B.8}$$

We consider the real-valued functions $G : \mathbb{R}^{3n \times 3n} \to \mathbb{R}$ and $g : \mathbb{R}^{n \times n} \to \mathbb{R}$ such that for any $\mathbf{L} \in \mathcal{B}_r(\mathbf{L}^*)$, we have the Taylor's expansion up to second order as follows,

$$G(\mathbf{L}) - G(\mathbf{L}^*) = \sum_{k=1}^2 g(\widehat{\mathbf{L}}_\gamma^{(k)}) - g(\mathbf{L}^{(k)*})$$

$$= \sum_{k=1}^2 \langle \nabla g(\mathbf{L}^{(k)*}), \Delta^{(k)} \rangle + \frac{1}{2} \sum_{k=1}^2 \operatorname{tr}\left(\operatorname{vec}(\Delta^{(k)})^\top \left[\nabla^2 g(\mathbf{L}^{(k)*})\right] \operatorname{vec}(\Delta^{(k)})\right)$$

$$\geq - \sum_{k=1}^2 \|\nabla g(\mathbf{L}^{(k)*})\|_F \|\Delta^{(k)}\|_F + \frac{1}{2} \sum_{k=1}^2 \lambda_{\min}\left(\nabla^2 g(\mathbf{L}^{(k)*})\right) \operatorname{tr}(\operatorname{vec}(\Delta^{(k)})^\top \operatorname{vec}(\Delta^{(k)})). \tag{B.9}$$

The last inequality follows from Lemmas B.2 and B.5 with $\Delta^{(1)} = \mathbf{L}^{(1)} - \mathbf{L}^{(1)*}$ and $\Delta^{(2)} = \mathbf{L}^{(2)} - \mathbf{L}^{(2)*}$. Now, $\nabla g(\mathbf{L}^{(k)*}) = 2\gamma_{1d}\left(\mathbf{L}^{(k)*} - \mathbf{I}_n \odot \mathbf{L}^{(k)*}\right) - \gamma_{2d}\left(\mathbf{I}_n \odot \mathbf{L}^{(k)*}\right)^{-1}$ and thus using

triangle inequality for Frobenius norm we have,

$$\sum_{k=1}^{2} \|\nabla g(\mathbf{L}^{(k)^*})\|_F \leq 2\gamma_{1d} \sum_{k=1}^{2} \left\| \mathbf{L}^{(k)^*} - \mathbf{I}_n \odot \mathbf{L}^{(k)^*} \right\|_F + \gamma_{2d} \sum_{k=1}^{2} \left\| \left( \mathbf{I} \odot \mathbf{L}^{(k)^*} \right)^{-1} \right\|_F$$

$$\leq 4\gamma_{1d} \max_{k \in \{1,2\}} \left\| \mathbf{L}^{(k)^*} - \mathbf{I} \odot \mathbf{L}^{(k)^*} \right\|_F + 2\gamma_{2d} \max_{k \in \{1,2\}} \left\| \left( \mathbf{I} \odot \mathbf{L}^{(k)^*} \right)^{-1} \right\|_F$$

$$\leq (4\gamma_{1d} + 2\gamma_{2d}) \max \left\{ \max_{k \in \{1,2\}} \left\| \mathbf{L}^{(k)^*} - \mathbf{I}_n \odot \mathbf{L}^{(k)^*} \right\|_F, \max_{k \in \{1,2\}} \left\| \left( \mathbf{I}_n \odot \mathbf{L}^{(k)^*} \right)^{-1} \right\|_F \right\}$$

$$= \tilde{\gamma}_d C_{\mathbf{L}} . \tag{B.10}$$

For the term consisting of the second derivative we have,

$$\sum_{k=1}^{2} \lambda_{\min} \left( \nabla^2 g(\mathbf{L}^{(k)^*}) \right) \mathrm{tr}(\Delta^{(k)^\top} \Delta^{(k)}) \geq \min_{k \in \{1,2\}} \lambda_{\min} \left( \nabla^2 g(\mathbf{L}^{(k)^*}) \right) \sum_{k=1}^{2} \mathrm{tr}(\mathrm{vec}(\Delta^{(k)})^\top \mathrm{vec}(\Delta^{(k)}))$$

$$\geq \frac{1}{C_r^2 r^2} \min_{k \in \{1,2\}} \lambda_{\min} \left( \nabla^2 g(\mathbf{L}^{(k)^*}) \right) \geq \frac{1}{C_r^2 r^2} \tilde{\lambda}_{\gamma_d} \|\Delta\|_F^2 \tag{B.11}$$

where $C_r$ is a constant with $C_r \geq \frac{1}{2K_r}$. Since both the inequalities B.10 and B.11 hold for any $\mathbf{L} \in \mathcal{B}_r(\mathbf{L}^*)$, we have he following,

$$G(\widehat{\mathbf{L}}_\gamma) - G(\mathbf{L}^*) \geq -\tilde{\gamma}_d C_{\mathbf{L}} \sum_{k=1}^{2} \|\Delta^{(k)}\|_F + \frac{1}{C_r^2 r^2} \tilde{\lambda}_{\gamma_d} \|\Delta\|_F^2 \geq -\tilde{\gamma}_d C_{\mathbf{L}} \|\Delta\|_F + \frac{1}{2C_r^2 r^2} \tilde{\lambda}_{\gamma_d} \|\Delta\|_F^2 . \tag{B.12}$$

For the penalizing term let us define $\mathcal{P} : \mathbb{R}^{3n \times 3n} \to \mathbb{R}$ and using lemma B.3 with $\mathbf{B} = \mathbf{I}_n$ we have,

$$\mathcal{P}(\mathbf{L}) - \mathcal{P}(\widehat{\mathbf{L}}_\gamma) = \gamma_{3d} \left( \|\mathbf{V}_\gamma\|_{2,1} - \|\widehat{\mathbf{V}}_\gamma\|_{2,1} \right)$$

$$\leq \gamma_{3d} \left\| \widehat{\mathbf{V}}_\gamma - \mathbf{V}_\gamma \right\|_{2,1}$$

$$\leq n^2 \gamma_{3d} \left\| \widehat{\mathbf{V}}_\gamma - \mathbf{V}_\gamma \right\|_F$$

$$\leq n^2 \gamma_{3d} \left\| \widehat{\mathbf{L}}_\gamma - \mathbf{L} \right\|_F . \tag{B.13}$$

Since the above relations hold for any $\mathbf{L} \in \mathcal{B}_r(\mathbf{L}^*) \in$, then it will also hold for $\mathbf{L}^*$. For the term dependent on the data using inequality in B.4 and lemmaB.6 with $t = \sqrt{d} - 2c^{(1)}\sqrt{n}$ and $t = \sqrt{d} - 2c^{(2)}\sqrt{n}$ respectively twice, we have,

$$\sum_{k=1}^{2} \mathrm{tr}\left( \left( \mathbf{L}^{(k)^*} - \widehat{\mathbf{L}}_\gamma^{(k)} \right) \left( \widehat{\Sigma}^{(k)} - \Sigma^{(k)^*} \right) \right) \leq \sum_{k=1}^{2} \left\| \widehat{\mathbf{L}}_\gamma^{(k)} - \mathbf{L}^{(k)^*} \right\|_F \left\| \widehat{\Sigma}^{(k)} - \Sigma^{(k)^*} \right\|_F$$

$$\leq \sqrt{d} \left\| \widehat{\mathbf{L}}_\gamma^{(k)} - \mathbf{L}^{(k)^*} \right\|_2 \left\| \widehat{\Sigma}^{(k)} - \Sigma^{(k)^*} \right\|_F$$

$$\leq \frac{n}{\sqrt{d}} \sum_{k=1}^{2} C^{(k)} \left\| \widehat{\mathbf{L}}_\gamma^{(k)} - \mathbf{L}^{(k)^*} \right\|_F$$

$$\leq \frac{2n}{\sqrt{d}} \max_{k \in \{1,2\}} C^{(k)} \left\| \widehat{\mathbf{L}}_\gamma - \mathbf{L}^* \right\|_F . \tag{B.14}$$

with probability at least $1 - 2 \left\{ \exp \left( -c^{(1)}(\sqrt{d} - 2c^{(1)}\sqrt{n})^2 \right) + \exp \left( -c^{(2)}(\sqrt{d} - 2c^{(2)}\sqrt{n})^2 \right) \right\}$ using Bonferroni's inequality on the probability of intersection of the events. Using lemma B.2 ,

$$\sum_{k=1}^{2} \mathrm{tr}((\widehat{\mathbf{L}}_\gamma^{(k)} - \mathbf{L}^{(k)^*})\Sigma^{(k)^*}) \geq -\sum_{k=1}^{2} \left\| \widehat{\mathbf{L}}_\gamma^{(k)} - \mathbf{L}^{(k)^*} \right\|_F \left\| \Sigma^{(k)^*} \right\|_F \geq -2 \left\| \widehat{\mathbf{L}}_\gamma - \mathbf{L}^* \right\|_F \max_{k \in \{1,2\}} \left\| \Sigma^{(k)^*} \right\|_F .$$

$$\tag{B.15}$$

Combining the above inequalities in B.8 we have,

$$-2\left\|\widehat{\mathbf{L}}_\gamma - \mathbf{L}^*\right\|_F \max_{k\in\{1,2\}}\left\|\Sigma^{(k)^*}\right\|_F - \tilde{\gamma}_d C_{\mathbf{L}}\left\|\widehat{\mathbf{L}}_\gamma - \mathbf{L}^*\right\|_F + \frac{1}{2r^2 C_r^2}\tilde{\lambda}_{\gamma_d}\left\|\widehat{\mathbf{L}}_\gamma - \mathbf{L}^*\right\|_F^2 \le \frac{2n}{\sqrt{d}}\max_{k\in\{1,2\}}C^{(k)}\left\|\widehat{\mathbf{L}}_\gamma - \mathbf{L}^*\right\|_F$$
$$+ n^2\gamma_{3d}\left\|\widehat{\mathbf{L}}_\gamma - \mathbf{L}\right\|_F. \tag{B.16}$$

Finally, we obtain,

$$\left\|\widehat{\mathbf{L}}_\gamma - \mathbf{L}^*\right\|_F \le \frac{4nr^2 C_r^2}{\tilde{\lambda}_{\gamma_d}\sqrt{d}}\max_{k\in\{1,2\}}C^{(k)} + \frac{2r^2 C_r^2}{\tilde{\lambda}_{\gamma_d}}\left\{n^2\gamma_{3d} + 2C_{\Sigma^*} + \tilde{\gamma}_d C_{\mathbf{L}}\right\}. \tag{B.17}$$

$\square$

### B.6 WELL-DEFINEDNESS OF THE PROJECTION WITH COERCIVE FUNCTION AND CLOSED SET

#### 1. SOME DEFINITIONS

**Coercive Function:** A function $f : \mathbb{R}^n \to \mathbb{R}$ is defined as **coercive** if it satisfies the condition:
$$\lim_{\|\boldsymbol{x}\|\to\infty} f(\boldsymbol{x}) = \infty.$$

In simpler terms, as the norm of $\boldsymbol{x}$ increases without bound, the value of $f(\boldsymbol{x})$ also tends to infinity. This indicates that the function's value becomes arbitrarily large as its input moves farther from the origin.

**Projection:** Given a point $\boldsymbol{x}' \in \mathbb{R}^n$, the **projection** onto a set $\mathcal{S}$ refers to finding the point $\boldsymbol{x}^* \in \mathcal{S}$ that has the shortest distance to $\boldsymbol{x}'$. Formally, the projection solves the following minimization problem:
$$\boldsymbol{x}^* = \arg\min_{\boldsymbol{x}\in\mathcal{S}} f(\boldsymbol{x}),$$
where $f(\boldsymbol{x}) = \|\boldsymbol{x} - \boldsymbol{x}'\|^2$ represents the squared Euclidean distance between $\boldsymbol{x}$ and $\boldsymbol{x}'$.

**Lemma B.9.** *The following projection*
$$\boldsymbol{x}^* = \arg\min_{\boldsymbol{x}\in\mathcal{S}} f(\boldsymbol{x}),$$
*is well-defined when $\mathcal{S}$ is closed and $f$ is coercive.*

*Proof.* Let's consider the following minimization problem:

$$\boldsymbol{x}^* = \arg\min_{\boldsymbol{x}\in\mathcal{S}} f(\boldsymbol{x}),$$

where $f : \mathbb{R}^n \to \mathbb{R}$ is a coercive function and $\mathcal{S}$ is a closed set. Because $f$ is coercive, for any sequence $\{\boldsymbol{x}_n\}$ where $\|\boldsymbol{x}_n\| \to \infty$, we have $f(\boldsymbol{x}_n) \to \infty$. This implies that the function grows without bound as $\boldsymbol{x}_n$ moves far from the origin. Consequently, the minimizer must be located in a **bounded region**.

Furthermore, since $\mathcal{S}$ is closed, any convergent sequence $\{\boldsymbol{x}_n\} \subseteq \mathcal{S}$ will have its limit point inside $\mathcal{S}$. This guarantees that there is at least one point within $\mathcal{S}$ that minimizes $f$. Hence, the minimizer $\boldsymbol{x}^*$ exists in $\mathcal{S}$, ensuring that the **projection is well-defined**.

$\square$

### ERROR BOUND IS PRESERVED FROM UNCONSTRAINED TO CONSTRAINED CASE USING PROJECTION

Let $\tilde{\mathbf{L}}^{(1)}$, $\tilde{\mathbf{L}}^{(2)}$, and $\tilde{\mathbf{V}}$ represent the unconstrained estimates of the true matrices $\mathbf{L}^{(1)^*}$, $\mathbf{L}^{(2)^*}$, and $\mathbf{V}^*$, respectively. In the unconstrained setting, we assume the following error bounds hold:
$$\|\tilde{\mathbf{L}}^{(1)} - \mathbf{L}^{(1)^*}\|_F \le \epsilon_1, \quad \|\tilde{\mathbf{L}}^{(2)} - \mathbf{L}^{(2)^*}\|_F \le \epsilon_2, \quad \|\tilde{\mathbf{V}} - \mathbf{V}^*\|_F \le \epsilon_V.$$
for some $\epsilon_1, \epsilon_2, \epsilon_V > 0$

PROJECTION ONTO $\mathbb{C} \cap \mathbb{L}$

The goal is to show that, after projecting onto the combined set $\mathbb{C} \cap \mathbb{L}$, the error bounds are preserved.

STEP-BY-STEP PROJECTION PROCESS

1. Projection onto Convex Set $\mathbb{L}$:

Since $\mathbb{L}$ is convex and closed, the projection onto this set is non-expansive with respect to the Frobenius norm. This means that for any matrix $\mathbf{X}$ and its projection $\mathbf{P}_{\mathbb{L}}(\mathbf{X})$ onto the set $\mathbb{L}$, the following inequality holds:

$$\|\mathbf{P}_{\mathbb{L}}(\mathbf{X}) - \mathbf{Y}\|_F \leq \|\mathbf{X} - \mathbf{Y}\|_F.$$

2. Projection onto Non-Convex Set $\mathbb{C}$:

The set $\mathbb{C}$ imposes a non-convex constraint on $\mathbf{L}^{(1)}$, $\mathbf{L}^{(2)}$, and $\mathbf{V}$. Even though this constraint is non-convex, the projection can still be non-expansive in a local region, which is typically achieved through methods like alternating minimization or proximal methods.

After projecting onto $\mathbb{C}$, the Frobenius norm of the error does not increase, which implies that:

$$\|\mathbf{P}_{\mathbb{C}}(\mathbf{P}_{\mathbb{L}}(\tilde{\mathbf{L}}^{(k)})) - \mathbf{L}^{(k)^*}\|_F \leq \|\mathbf{P}_{\mathbb{L}}(\tilde{\mathbf{L}}^{(k)}) - \mathbf{L}^{(k)^*}\|_F,$$

for $k = 1, 2$, and similarly for $\mathbf{V}$.

3. Combined Projection onto $\mathbb{C} \cap \mathbb{L}$:

The combined projection first projects onto $\mathbb{L}$ (convex) and then onto $\mathbb{C}$ (non-convex). Since both projections are non-expansive, the total error after the projection remains bounded. Therefore, we conclude that:

$$\|\mathbf{P}_{\mathbb{C} \cap \mathbb{L}}(\tilde{\mathbf{L}}^{(1)}) - \mathbf{L}^{(1)^*}\|_F \leq \|\tilde{\mathbf{L}}^{(1)} - \mathbf{L}^{(1)^*}\|_F \leq \epsilon_1,$$
$$\|\mathbf{P}_{\mathbb{C} \cap \mathbb{L}}(\tilde{\mathbf{L}}^{(2)}) - \mathbf{L}^{(2)^*}\|_F \leq \|\tilde{\mathbf{L}}^{(2)} - \mathbf{L}^{(2)^*}\|_F \leq \epsilon_2,$$
$$\|\mathbf{P}_{\mathbb{C} \cap \mathbb{L}}(\tilde{\mathbf{V}}) - \mathbf{V}^*\|_F \leq \|\tilde{\mathbf{V}} - \mathbf{V}^*\|_F \leq \epsilon_V.$$

CONCLUSION

After applying the combined projection onto the set $\mathbb{C} \cap \mathbb{L}$, the error bounds for the matrices $\mathbf{L}^{(1)}$, $\mathbf{L}^{(2)}$, and $\mathbf{V}$ are preserved.

## C  RESULTS FOR RGG

Figure 5 presents the results for the RGG network using three types of graph filters, varying the number of samples, perturbed nodes, and noise levels. As the number of signal samples increases, all methods show improvement, with PN-TVL achieving the best results. However, the performance declines as the number of perturbed nodes and noise increases. While MVGL is resilient to noise, it becomes less effective with more perturbed nodes due to its emphasis on maximizing edge similarity. In particular, its performance becomes worse than PNJGL and SV. PNJGL outperforms MVGL in these scenarios by assuming node connectivity drives view differences. It is also interesting to note that for RGG networks PNGL's performance deteriorates for non-Gaussian signal models (rows 2 and 3) compared to the Gaussian signal model (row 1). In particular, while PNJGL performs better than SV for increasing number of perturbed nodes in the Gaussian case, it performs worse for the non-Gaussian case. This illustrates the limitations of GMMs for graph inference.

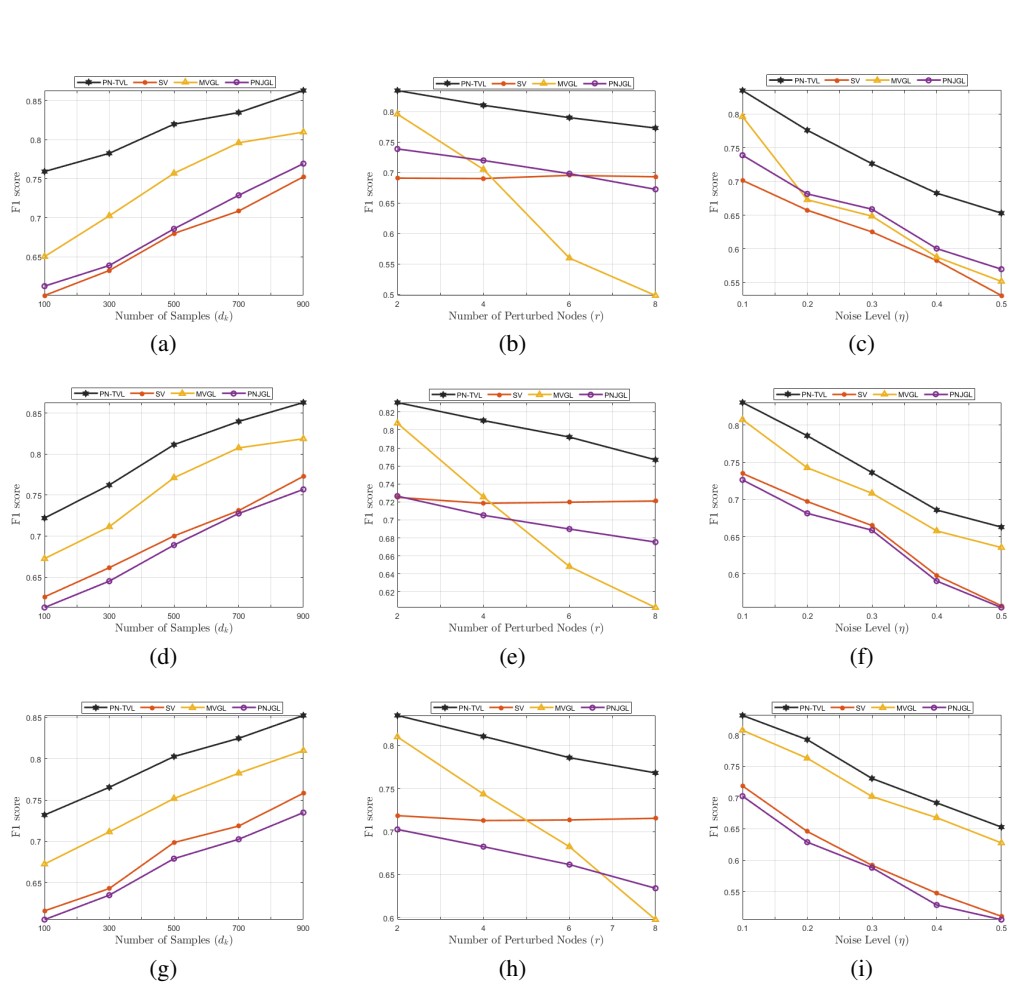

Figure 5: Comparison of performance for RGG network model with three different graph filters for varying signal parameters: (a)-(c) varying number of samples $(d_k)$, number of perturbed nodes $(r)$, and noise level $(\eta)$, for Gaussian filter, (d)-(f) varying number of samples $(d_k)$, number of perturbed nodes $(r)$, and noise level $(\eta)$, for heat filter, (g)-(i) varying number of samples $(d_k)$, number of perturbed nodes $(r)$, and noise level $(\eta)$, for Tikhonov filter.

