# OpenReview forum: "Node-based Multiple Graph Learning with Theoretical Guarantees"
_ICLR.cc/2025/Conference — Submitted to ICLR 2025_

### Official Review · Reviewer_EeY6 · 2024-10-27

**Soundness:** 3
**Presentation:** 2
**Contribution:** 2
**Rating:** 3
**Confidence:** 3

**Summary:**

This paper presents a graph signal processing method to learn the graph Laplacians of two graphs under the assumption that graph signals are smooth and that variations between the two views arise from perturbations in node connectivity. The authors explicitly design a model employing an ADMM solver and provide a theoretical analysis of the error bound. A notable result is that increasing the sample size alone does not guarantee convergence to the true values. Instead, convergence also depends critically on the topology of the underlying true graph structure.

**Strengths:**

1. Compared to existing techniques that focus on edge-similarity based graph learning, this paper investigates node perturbation.

2. This paper characterizes the graph via Laplacian matrix, rather than following previous Gaussian Graphical Models.

3. This paper utilizes ADMM optimization, which is interpretable and provides the upper bound on the estimation error.

**Weaknesses:**

1. While the paper provides interesting insights, it appears to overclaim its contributions. The focus is on node perturbation in multi-view graph learning; however, the method merely offers solution for two-view cases. Such a claim lacks rigor in the context of research.

2. The components of the proposed model in Equation 4.1 seem to be a combination of existing techniques. For instance, the first term quantifies the variation of graph signals (backbone), while the second term can be simplified to $\|\|-\mathbf{W}^{(k)}\|\|_{F}^{2}$ (notably, $\mathbf{L} = \mathbf{D} - \mathbf{W}$), which regulates the sparsity of the affinity graph through the parameter $\gamma_1$. The third term introduces a penalty to avoid zero values in the elements of the degree matrix using $\log(\cdot)$ (Kalofolias et al., 2017), and the fourth term resembles techniques from RCON (Mohan et al., 2014). Consequently, the formulation appears to be less novel and more a synthesis of existing methods.

3. The paper does not provide any analysis of computational complexity. In fact, the method is with high computational and space complexities, which limits its applicability in large-scale scenarios.

**Questions:**

In addition to my concerns in Weaknesses, I have the following comments:

1. The authors should provide a careful summary of their contributions, particularly since the model appears to be a combination of existing techniques. Furthermore, the ADMM solver seems straightforward and lacks novel insights. While the derived error bound appears solid, I question how this bound will guide future research. Is there any difference between this error bound and previous research?

2. Could you generalize your model into multi-view version? It seems that the current paper lacks the completeness necessary to substantiate claims of addressing node perturbation in multi-view graph learning.

3. The baseline method, MVGL, supports multiple views, whereas the proposed model is limited to handling only two. What advantages does your approach offer over MVGL?

4. Do you have any recommendations for initializing the ADMM solver? Have you identified any initialization methods that perform better than others in your experiments? Additionally, I suggest including the convergence curves.

---

### Official Review · Reviewer_bAAL · 2024-11-03

**Soundness:** 3
**Presentation:** 3
**Contribution:** 2
**Rating:** 5
**Confidence:** 4

**Summary:**

This paper is concerned with the problem of inferring multiple related
graphs from nodal observations. In a graph signal processing (GSP)
framework, these are considered as graph signals, which are
incorporated in their optimization program via a smoothness penalty
when inferring the graph structure.

The novelty of this work comes primarily from the use of a mixed-norm
penalty in the two-view graph learning problem, which draws
inspiration from the RCON penalty of Mohan et al. (2014). The
difference in this work, however, is the general smoothness assumption
on the graph signals, as opposed to the particular case of a gaussian
graphical model. Some guarantees are given (Theorem 5.1), as well as
empirical evidence for the utility of the proposed routine.

**Strengths:**

This paper provides a straightforward approach to a well-defined
problem, with reasonable algorithms and empirical evidence to back up
the utility of the proposed approach. I can imagine the proposed
method being useful to practitioners, particularly if they find the
results of optimization programs based on a gaussian graphical model
assumption to have subpar outputs.

**Weaknesses:**

The novelty of this work is limited -- it seems to be a combination of
penalties used in the GSP and GGM literature, brought together for a
new special case. Although I see the utility of using the RCON penalty
outside of the context of GGMs, the intellectual contribution of this
paper is incremental, not offering much more insight than another tool
in the toolbox. I do not want to downplay the utility of such
contributions, but it is nonetheless a weakness of this work.

I also have a major concern about the theoretical result in Theorem
5.1, and its relation to Assumptions A1-A3. It is not clear how the
signals relate to the graphs at all under these assumptions. Implicit
in the penalty is an assumption that the signals are smooth, but I
don't see this clearly stated anywhere in the assumptions. Based on my
reading of the assumptions, I could very well apply the result to
signals support in a "high-pass" band of the spectrum of the graphs
and not expect much difference in the recovery performance. I am not
sure if I am right here, but the results of Theorem 5.1 don't seem to
describe much in this regard, given that they do not (seem to!) rely
on the relationship between the graphs and the signals. This is
concerning, as the whole point of the paper is to infer graph
structure from node signals that are assumed to have some relationship
to the graph structure.

Of course, under the reasonable assumption that the signals are smooth
with respect to the graphs that they are supported on, it makes sense
that the proposed algorithm works -- which is the setting of all of
the generating graph filters described in Section 6.1.1.

**Questions:**

Please clarify how the results of Theorem 5.1 depend on the graph signal structure, i.e., on being low-pass or something to that effect -- otherwise, please clarify the utility of this theorem if the result does not depend on that property.

---

### Official Review · Reviewer_fnog · 2024-11-12

**Soundness:** 2
**Presentation:** 3
**Contribution:** 2
**Rating:** 3
**Confidence:** 4

**Summary:**

This paper furthers the area of  learning graph structure from multiple views, a problem important in Graph Signal Processing(GSP). It proposes a formulation(4.1) for learning the Laplacian for each view. The formulation is solved through ADMM and guarantees are given.

**Strengths:**

Following are some of the strengths of the paper
- It is rigorous in the formulation and the algorithmic guarantees it provide
- It provides Node based learning, in contrast to edge based learning which is the state of the art
- it is easy to follow.

**Weaknesses:**

The paper appears to have several shortcomings.
- Contributions seem to be incremental.
-Assumptions look a little restrictive.
- It is restricted only to two views and not multiple views
- Insights from Analysis is missing
Few suggestions are as follows
-A more comprehensive discussion contrasting the formulations(4.1 and 4.3) with existing formulations and highlighting the relative advantages will be useful.
-Is it possible to quote a result from literature which shows how standard the assumptions are and what are the corresponding guarantees.
-If the formulation is specialised to two-views then the paper should be renamed as such.
-To understand the importance of the derived results it would be good to add key insights. For example if
d more than n^2 then the learning is good.

**Questions:**

The following clarifications will help in addressing some of the weakness

- Formulation 4.1

   - What is the need for adding sparsity? It requires $\gamma_2$. If it is not added how will the formulation and analysis look like.

   - is it the case that the formulation is restricted to two views because you need $L_1-L_2 = V + V^\top$

- Assumptions
  - Can they be contrasted more with state of the art. For example if the signals are gaussian, what is the result already known and how does it compare with the subgaussian result discussed here . This will help explain the power of the results obtained. Good to quote results from Gaussian Graphical models to show the importance.

- Analysis
  - Provide some insights from the analysis. Can we say that if $ d \in \Omega(n^2)$ only then the learning will be good(the first term suggests so). If this is indeed the case does the experimental result suggest that
  - Are there any regimes for the hyper parameters for which the learning is good. Just saying they are positive may not yield comprehensive understanding.

---

### Meta-Review · Area_Chair_BXfd · 2024-12-17

**Metareview:**

The paper aims to address the problem of learning edges from multivew graph information. The results seem to be limited to two graphs and also under restrictive assumptions. Unfortunately the authors did not submit any rebuttal. Thus, keeping in mind the concerns, the recommendation is reject.

**Additional Comments On Reviewer Discussion:**

No rebuttal was submitted. There were several concerns and they are not addressed.

---

### Decision · Program_Chairs · 2025-01-22

Reject